# Membrane-mediated interaction of non-conventional snake three-finger toxins with nicotinic acetylcholine receptors

Zakhar O. Shenkarev [1,2,8], Yuri M. Chesnokov[3,4,8], Maxim M. Zaigraev [1,2,8], Anton O. Chugunov [1,2,5,8], Dmitrii S. Kulbatskii[1,8], Milita V. Kocharovskaya[1,2], Alexander S. Paramonov[1], Maxim L. Bychkov[1], Mikhail A. Shulepko[1], Dmitry E. Nolde [1,5], Roman A. Kamyshinsky[3,4], Evgeniy O. Yablokov [6], Alexey S. Ivanov [6], Mikhail P. Kirpichnikov[1,7] & Ekaterina N. Lyukmanova [1,2,7 ✉]

Nicotinic acetylcholine receptor of α7 type (α7-nAChR) presented in the nervous and immune systems and epithelium is a promising therapeutic target for cognitive disfunctions and cancer treatment. Weak toxin from *Naja kaouthia* venom (WTX) is a non-conventional three-finger neurotoxin, targeting α7-nAChR with weak affinity. There are no data on interaction mode of non-conventional neurotoxins with nAChRs. Using α-bungarotoxin (classical three-finger neurotoxin with high affinity to α7-nAChR), we showed applicability of cryo-EM to study complexes of α7-nAChR extracellular ligand-binding domain (α7-ECD) with toxins. Using cryo-EM structure of the α7-ECD/WTX complex, together with NMR data on membrane active site in the WTX molecule and mutagenesis data, we reconstruct the structure of α7-nAChR/WTX complex in the membrane environment. WTX interacts at the entrance to the orthosteric site located at the receptor intersubunit interface and simultaneously forms the contacts with the membrane surface. WTX interaction mode with α7-nAChR significantly differs from α-bungarotoxin's one, which does not contact the membrane. Our study reveals the important role of the membrane for interaction of non-conventional neurotoxins with the nicotinic receptors.

[1] Shemyakin-Ovchinnikov Institute of Bioorganic Chemistry, Russian Academy of Sciences, Miklukho-Maklaya 16/10, Moscow 117997, Russia. [2] Phystech School of Biological and Medical Physics, Moscow Institute of Physics and Technology (National Research University), Institutsky Lane 9, Dolgoprudny, Moscow 141701, Russia. [3] National Research Center "Kurchatov Institute", Academic Kurchatov Sq. 1, Moscow 123182, Russia. [4] Shubnikov Institute of Crystallography of Federal Scientific Research Centre "Crystallography and Photonics" of Russian Academy of Sciences, Leninsky Prospect 59, Moscow 119333, Russia. [5] National Research University Higher School of Economics, Myasnitskaya Str. 20, Moscow 101000, Russia. [6] Institute of Biomedical Chemistry, Pogodinskaya 10k8, Moscow 119121, Russia. [7] Interdisciplinary Scientific and Educational School of Moscow University "Molecular Technologies of the Living Systems and Synthetic Biology", Faculty of Biology, Lomonosov Moscow State University, Leninskie Gory, Moscow 119234, Russia. [8]These authors contributed equally: Zakhar O. Shenkarev, Yuri M. Chesnokov, Maxim M. Zaigraev, Anton O. Chugunov, Dmitrii S. Kulbatskii. ✉email: ekaterina-lyukmanova@yandex.ru

Nicotinic acetylcholine receptors (nAChRs) are ligand-gated ion channels, involved in a variety of important physiological processes like modulation of behavior and memory[1], neuromuscular transmission[2], control of epithelial cells homeostasis[3,4] and many others. The α7 type nAChR (α7-nAChR) is of special interest, because this receptor is a promising therapeutic target for treatment of cognitive disorders associated with Alzheimer disease, schizophrenia, and depression[5–7] and cancer[8].

α7-nAChR is a homopentameric membrane protein consisting of an extracellular ligand-binding domain (α7-ECD), a trans-membrane (TM) domain responsible for ionic conduction, and an intracellular domain with a large and flexible intracellular loop[9]. α7-nAChR has multiple functional states[10] and can interact with other receptors[11,12], intracellular kinases[11], and, probably, with G-proteins[13]. An important breakthrough in the structural studies of α7-nAChR interactions with the ligands was made using α7-ECD chimera with acetylcholine-binding protein (AChBP) from *Lymnaea stagnalis*[14]. This chimeric protein has a 71% sequence similarity with the human α7-nAChR extracellular domain and was used for X-ray studies of the ligand-receptor complexes[14,15]. Recently, structures of α7-nAChR with a modified intracellular loop in a complex with different ligands have been determined by cryo-EM[16,17].

Proteins from the Ly6/uPAR family have a characteristic fold with three loops (fingers) protruding from a β-structural core, stabilized by a system of four invariant disulfide bonds[18]. The most extensively studied three-finger proteins are α-neurotoxins from snake venom acting on different subtypes of nAChRs[19]. There are three types of snake three-finger α-neurotoxins: short-chain, long-chain, and non-conventional. For example, α-bungarotoxin from *Bungarus multicinctus* (α-Bgtx) is the long-chain neurotoxin and inhibits muscle-type and α7-type nAChRs with affinity in the nanomolar range[20–22]. Unlike to short-chain and long-chain neurotoxins, the non-conventional neurotoxins have an additional disulfide bond in the loop I[18,19]. The non-conventional neurotoxins can be classified into strong and weak toxins. A typical representative of the latter group is the weak toxin from *Naja kaouthia* venom (WTX), which demonstrates a low affinity both for muscle-type and α7-type nAChRs in the range of tens of μM[23–25]. To date, the structures of short-chain and long-chain neurotoxins in complex with muscle-type or α7-type nAChRs were determined by cryo-EM[16,26,27]. In contrast, no structures of the complexes of non-conventional toxins with nAChRs are known to date.

Here, using α-Bgtx, we showed the ability to use cryo-EM for studies of α7-ECD in complex with three-finger neurotoxins. At the same time, the cryo-EM study of the α7-ECD/WTX complex was complicated by preferred orientation of the particles. Therefore, the initial cryo-EM model was refined using NMR data on the WTX membrane active site and mutagenesis data on the role of different toxin's residues in the interaction with a lipid membrane and full-length α7-nAChR. As a result, we reconstructed the structure of the complex of full-length α7-nAChR with WTX in the membrane environment. This work revealed the molecular mechanisms underlying the action of weak non-conventional neurotoxins and showed the importance of the membrane environment in their binding to the receptor.

## Results

**Cryo-EM study of α7-ECD/α-Bgtx complex.** To assess the principal possibility of using cryo-EM to study α7-ECD complexes with three-finger proteins, long-chain toxin α-Bgtx (Supplementary Fig. 1) was chosen. The known X-ray structure of the α7-ECD/α-Bgtx complex was used here as a reference[15].

### Table 1 CryoEM data collection and processing.

| Microscope | α7-ECD/α-Bgtx dataset | α7-ECD/WTX dataset |
|---|---|---|
| | Titan Krios (Thermo Fisher Scientific) | Titan Krios (Thermo Fisher Scientific) |
| Camera | Falcon II | Falcon II |
| Voltage (kV) | 300 | 300 |
| Magnification | 75,000 | 75,000 |
| Number of micrographs | 2245 | 1981 |
| Total dose (e$^-$/Å$^2$) | 80 | 80 |
| Dose per frame (e$^-$/Å$^2$) | 2 | 2.7 |
| Number of frames | 40 | 30 |
| Total exposure time (s) | 2 | 1.5 |
| Defocus range (μm) | 0.6–2.5 | 1.0–3.0 |
| Pixel size (Å) | 0.86 | 0.86 |
| Symmetry imposed | C5 | C5 |
| Initial number of particles | 1,517,534 | 703,453 |
| Final number of particles | 27,813 | 108,680 |
| Map resolution (Å) | 3.37 | 5.61 |
| FSC threshold | 0.143 | 0.143 |

Initial cryo-EM screening of α7-ECD alone as well as in complex with α-Bgtx using Quantifoil R1.2/1.3 Cu grids revealed that the most of the particles demonstrated preferred top-view orientation. To overcome this issue, we tested the addition of different detergents in low concentrations (CHAPSO, LMNG, TWEEN-20, DDM, Fos-Choline8 Fluorinated (F-Fos-Ch8)), that in some cases help to solve the problem with the preferred orientation[28]. The side-view projections of the α7-ECD/α-Bgtx complex were observed only in the case of F-Fos-Ch8, but the image quality deteriorated significantly with increase of the detergent concentration (from 0.3 to 3 mM). Thereafter, we tested gold grids (Quantifoil R1.2/1.3 Au and GO on Quantifoil R1.2/1.3 coated with graphene oxide support film), and the last ones provided the side-view projections of the α7-ECD/α-Bgtx complex on the micrographs.

2D classification of the α7-ECD/α-Bgtx images showed five symmetrically positioned α7-ECD subunits with five additional prominent densities at the interfaces of adjacent subunits (Fig. 1a). These additional densities, corresponding to the α-Bgtx molecules, were absent in the images of α7-ECD without the added toxin (Fig. 1a). The secondary structure of α7-ECD subunits was clearly visible in the images. About 80% of the particles belonged to the 2D classes with the top-view orientation, while ~20% of the particles were tilted (Fig. 1a and Supplementary Fig. 2). After 2D and 3D classification, the final 3D refinement of a cryo-EM map was done with C5 symmetry (Table 1 and Supplementary Fig. 3a). Resolution of the obtained 3D structure, estimated according to the "gold-standard" criterion (FSC = 0.143), was about 3.4 Å (Fig. 1b, c and Supplementary Fig. 3d). The local resolution in the region of α7-ECD was higher (~3 Å) than in the toxin region (~4 Å). This illustrates the motions of the α-Bgtx molecules within the complex. The distribution analysis of orientations showed that the most of the "side-view" particles are tilted 25–35° relative to the C5 symmetry axis (Fig. 1a and Supplementary Fig. 2).

In the obtained structure, α-Bgtx binds to the center of the cylindrical outer wall of the receptor domain, so the toxin locates far from the top and bottom sides of the ECD and far from the expected position of the lipid membrane surrounding full-length α7-nAChR (Fig. 1b, c). The plane defined by the loops and β-sheets of the α-Bgtx molecule is tilted by −45° relative to the C5 symmetry axis of the receptor domain (Fig. 1b, c, negative sign

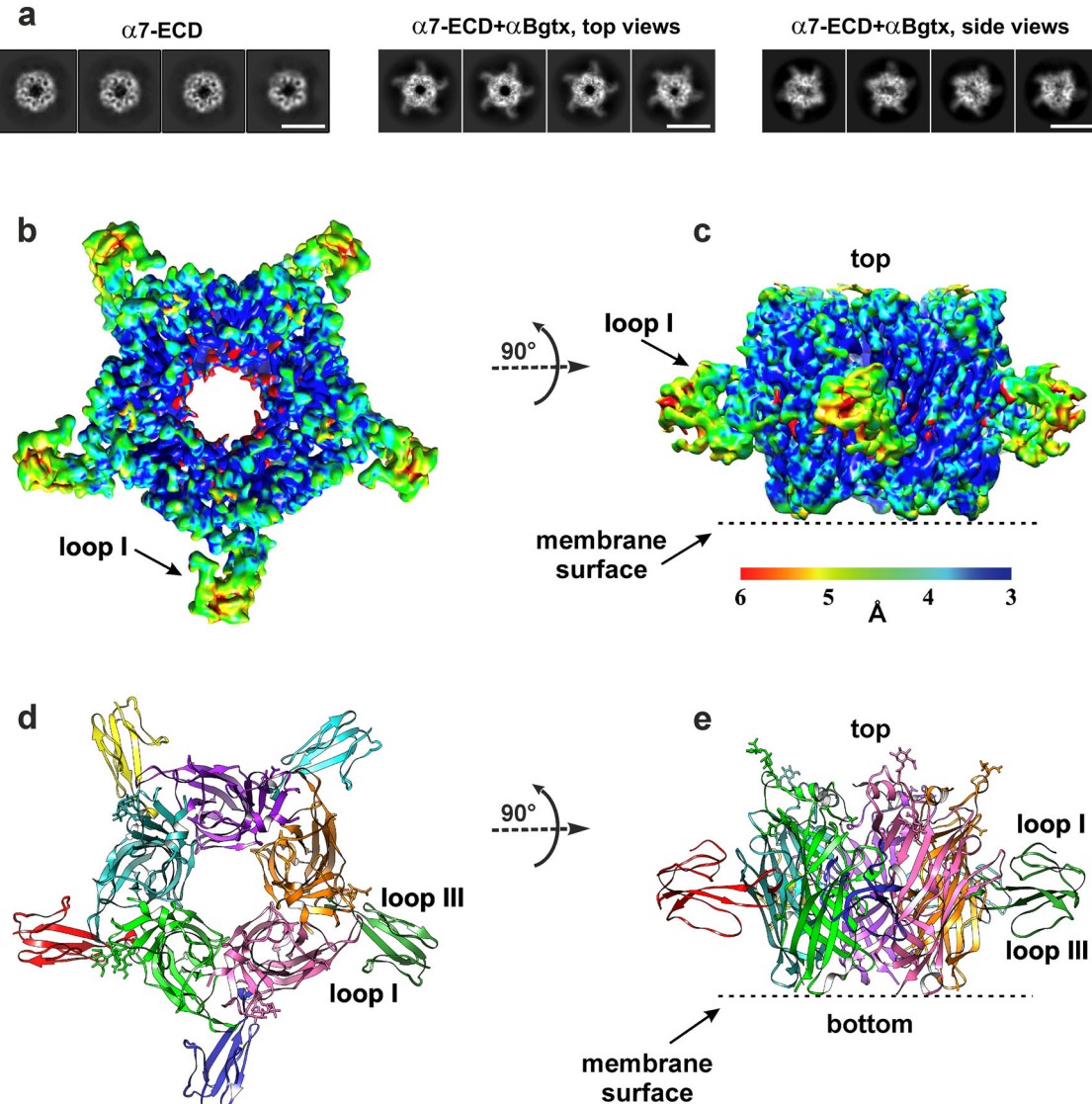

**Fig. 1 3D reconstruction of α7-ECD/α-Bgtx complex determined by cryo-EM. a** Representative 2D class averages for α7-ECD without and with α-Bgtx. Scalebar is 10 nm. **b, c** Final 3D density map of the α7-ECD/α-Bgtx complex colored with local resolution in top and side views. **d, e** Crystal structure of the α7-ECD/α-Bgtx complex (PDB ID: 4HQP[15]) is shown for comparison in the same orientation as **b, c**. Five subunits of α7-ECD and five α-Bgtx molecules are differently colored. The position of the toxin's loops I and III and expected position of membrane interface in the full-length channel are shown.

corresponds to anti-clockwise rotation). The loop I of α-Bgtx, clearly visible on the cryo-EM map as the individual density, is the furthest from the expected membrane surface (Fig. 1b). Comparison with the known crystal structure of the α7-ECD/α-Bgtx complex (PDB ID: 4HQP[15], Fig. 1d, e and Supplementary Fig. 4) showed no significant differences from the cryo-EM map. The cryo-EM map allowed to identify the backbone and amino acid side chains in the α-helices and β-strands of ECD (Supplementary Fig. 4). Toxin's β-strands were visible too, but the resolution is not enough to visualize the flexible toxin's loops.

**Cryo-EM study of the α7-ECD/WTX complex.** Analysis of the cryo-EM images of the α7-ECD complex with WTX revealed that almost all particles had the preferred top-view orientation (Supplementary Fig. 5a). 2D classification demonstrated the clearly visible secondary structure of five α7-ECD subunits and five additional symmetrically positioned weak densities absent in the classes of α7-ECD without the added toxin (Fig. 2a and Supplementary Fig. 5b). We suggested that WTX, similarly to α-Bgtx, binds to α7-ECD at five symmetrical sites at the intersubunit

interfaces. Contrarily to the case with α-Bgtx, addition of detergents and variation in the type of grids did not provide the side-view orientations of the α7-ECD/WTX particles. Therefore, we collected micrographs at 15° and 30° tilt angles (Supplementary Fig. 5b). The tilting of grids decreased the contrast, so the fine features of the images became poorly visible. The final 3D refinement was done using C5 symmetry (Table 1 and Supplementary Fig. 5c). Global resolution of the obtained 3D map estimated by the "gold-standard" criterion was 5.61 Å (Supplementary Fig. 5d). However, the visual inspection did not reliably support the estimated result since the typical high-resolution features were not observed due to a resolution anisotropy (Fig. 2b). Most of the high-resolution data came from the particles obtained at 0° tilt, which caused overfitting. The other reason for low resolution obtained could be a high mobility of the bound WTX molecules.

Five distinct α7-ECD subunits and their orientation can be clearly identified on the obtained 3D map (Fig. 2b). The top side of α7-ECD (relative to the expected membrane interface in the full-length receptor) showed the bulges corresponding to the glycan fragments linked to the N66 side-chain (Fig. 2b, arrows).

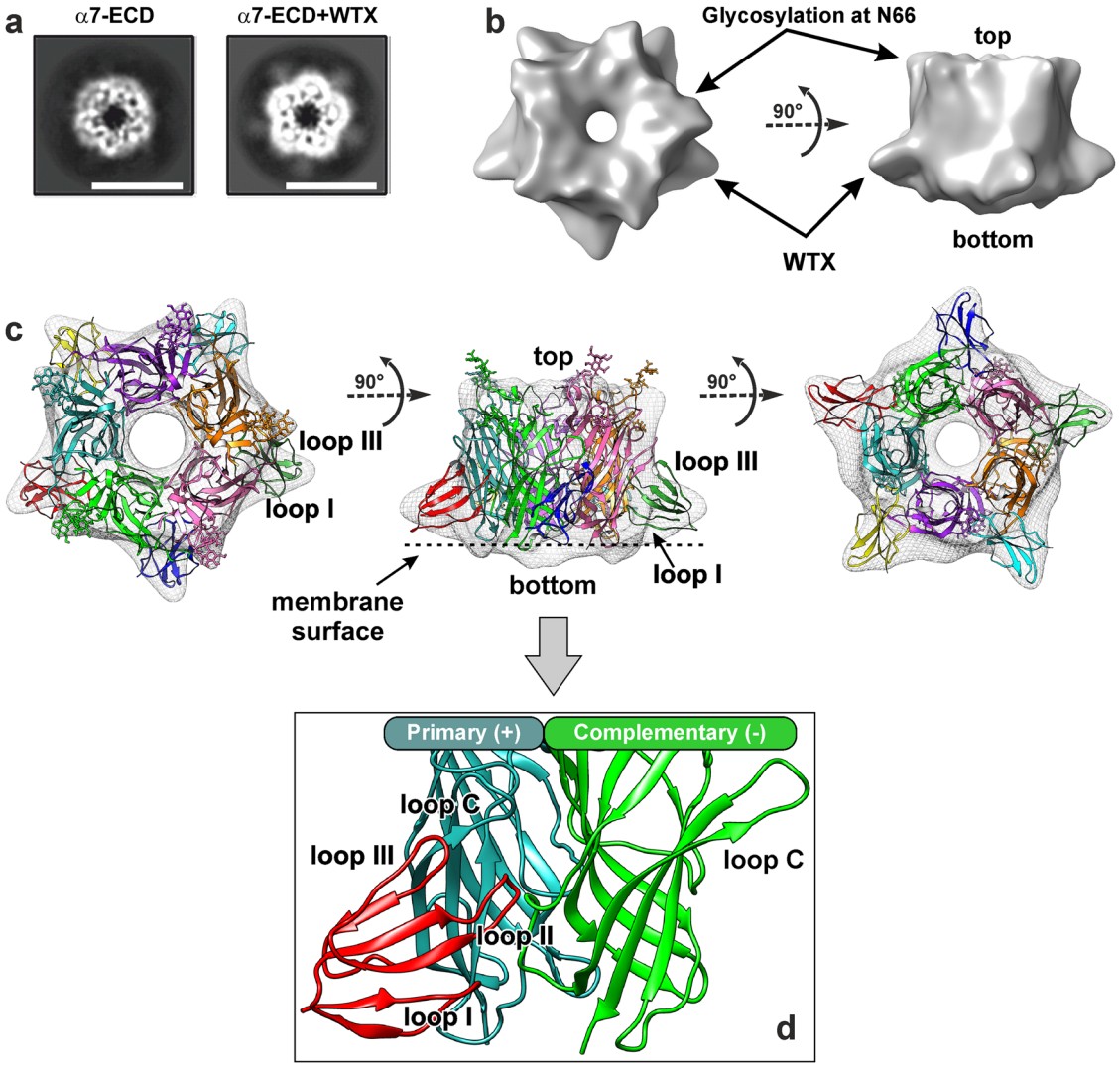

**Fig. 2 3D reconstruction of α7-ECD/WTX complex determined by cryo-EM and computer modeling. a** Comparison of the representative top-view 2D class averages for α7-ECD and α7-ECD/WTX complex. The fine features of pentameric domain complex and density corresponding to five WTX molecules are visible. Scalebar is 10 nm. **b** 3D density map of α7-ECD/WTX complex. The features corresponding to the linked glycan fragments and bound WTX molecules are indicated. **c** Model of α7-ECD/WTX complex fitted into experimental 3D density map. The position of the toxin's loops I and III and expected position of membrane interface in the full-length channel are shown. **d** Enlarged view of WTX interaction with the orthosteric ligand-binding site of α7-ECD located under the loop C of the primary (+) receptor subunit at the interface between the primary (+) and complimentary (−) subunits.

In contrast, the large densities corresponding to five bound WTX molecules were identified close to the bottom side of α7-ECD near the expected surface of the membrane surrounding the full-length receptor. The overall shape of these densities was in good agreement with the shape expected for the three-finger molecule, which interacts by its loops with the interface of the adjacent α7-ECD subunits. In contrast to the situation observed for α-Bgtx, the plane defined by the loops and β-sheets of the WTX molecule was ∼ +35° tilted relative to the C5 symmetry axis of ECD (Fig. 2b, positive sign corresponds to clockwise rotation). Two orientations of the three-finger WTX molecule within the corresponding density were possible: with the loop I or loop III of the toxin directed toward the membrane. In the both cases the toxin's loop II was directed toward the intersubunit interface of ECD.

**NMR revealed membrane active site in the WTX molecule.** We hypothesized that if WTX interacts with the "lower" part of α7-ECD, then it can contact the membrane environment of full-length α7-nAChR. In line with this assumption, we observed the toxin binding to small unilamellar lipid vesicles (liposomes) containing a mixture of zwitterionic lipid (palmitoyl-oleoyl-phosphatidylcholine, POPC), anionic lipid (palmitoyl-oleoyl-phosphatidylglycerol, POPG), and cholesterol (CHOL). Intensity of NMR signals of WTX decreased upon addition of the vesicles (Fig. 3a) due to association of the toxin molecules with the lipid membrane. To determine the membrane binding site in the WTX molecule, we minimized a concentration of the charged lipids in the vesicles (POPC:POPG:CHOL = 7:1:2) and shielded electrostatic interactions by 50 mM NaCl. In this case the differential attenuation of the WTX signals in the $^{15}$N-HSQC spectrum was observed. The signals of the residues that participate in the membrane binding were attenuated more strongly than the signals of non-participating residues (Fig. 3b). Obtained data revealed two possible membrane-binding sites in the WTX molecule formed by: (1) the β-sheet in the loop I, the neighboring "head-1" region, and the N- and C-terminal regions, and (2) the β-sheet in the loops II and III and the tip of loop II (Fig. 3c, blue

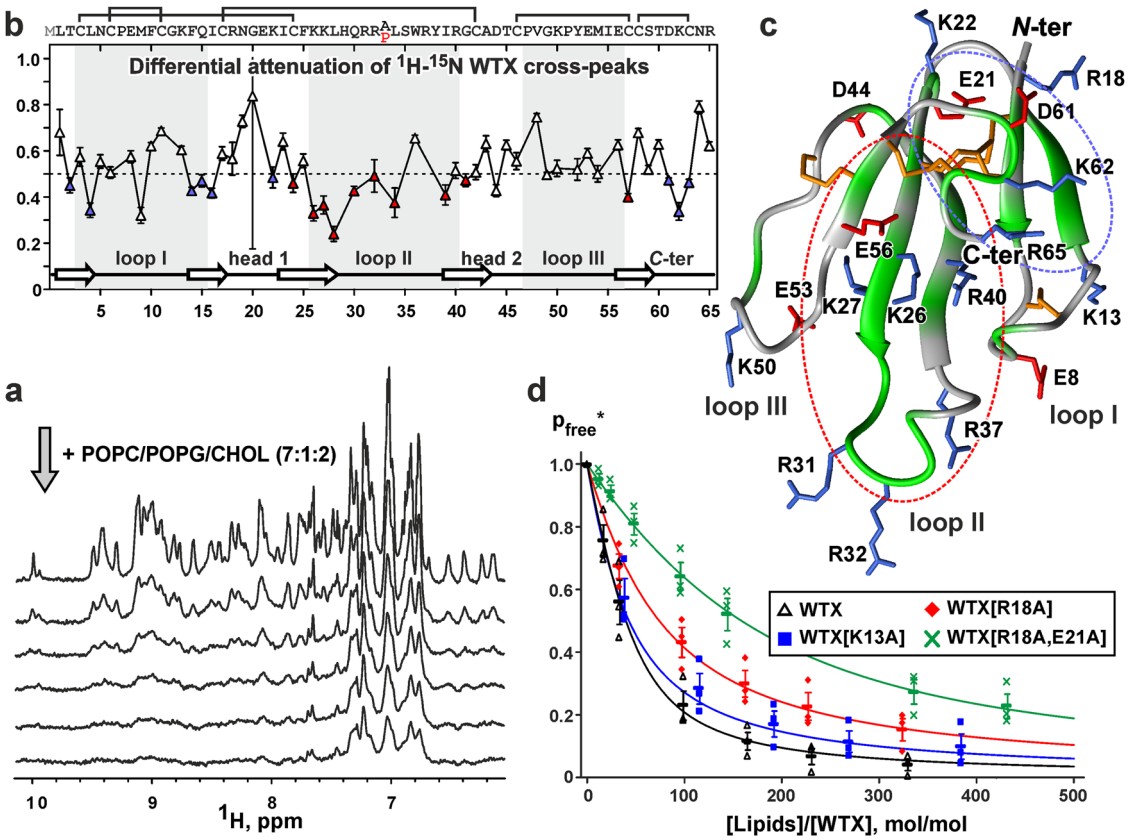

**Fig. 3 Interaction of WTX and its mutants with lipid vesicles. a** Amide-aromatic regions of 1D NMR spectra of WTX measured at different vesicle concentrations. The drop in intensity of NMR signals corresponds to binding of the toxin to the vesicles and decrease in the apparent fraction of the free toxin in solution ($P_{free}*$). **b** Relative attenuation of intensities of backbone NH cross-peaks in the $^{15}N$-HSQC spectrum of WTX induced by addition of POPC/POPG/CHOL vesicles. Data are values obtained from a single NMR experiment. Error bars are experimental errors. The 0.5 threshold line subdivides the data in two groups: the residues interacting with the vesicles or not. Color code corresponds to the regions highlighted by ellipses on **c**. **c** Spatial structure of WTX[P33A] mutant (used for the model building, see Methods) in ribbon representation. Disulfide bonds, positively charged, and negatively charged residues are colored in orange, blue, and red, respectively. The protein ribbon is colored according to vesicle-binding data from **b**. The residues interacting with the membrane are in green. Two protein regions responsible for protein-membrane interaction are shown by dashed ellipses. **d** Binding isotherms for WTX and its mutants are approximated by the Langmuir equation. Fitted parameters are summarized in Supplementary Table 3. The $P_{free}$ values shown are the mean ± S.E.M. of three independent titration experiments ($n = 3$). *$p < 0.05$ and ***$p < 0.001$ indicate the significant difference between groups by Repeated-Measures one-way ANOVA/Tukey's test. Please note that obtained isotherms represent upper approximation to the real binding parameters. In the presence of slow-intermediate (on NMR timescale) exchange between the free toxin in solution and membrane bound toxin, the apparent $P_{free}*$ value calculated from the NMR signals intensity is always lower than true $P_{free}$ value.

and red ellipses, respectively). Both sites contained a high amount of positively charged residues (K13, R18, K62, and R65 in the first site, and K26, K27, R31, R32, R37, and R40 in the second one). No strong attenuation of the NMR signals was observed in the loop III of WTX (Fig. 3b). Thus, the toxin tends to bind the membrane with the loops I or II, but not with the loop III. However, according to the cryo-EM data (Fig. 2), the loop II is directed toward the intersubunit interface of α7-ECD and cannot interact with the membrane. So, one orientation of the WTX molecule in the complex with α7-ECD is possible, when the loop I is directed toward the expected surface of the lipid membrane surrounding the receptor (Fig. 2c, d).

**Integration of cryo-EM, NMR, and mutagenesis data with in silico modeling yielded the structure of full-length α7-nAChR/ WTX complex.** Low resolution of the obtained cryo-EM map did not permit unambiguous reconstruction of the spatial structure of the α7-ECD/WTX complex; thus, we combined previous mutagenesis data[24] and NMR-derived orientation restraints with the fitting of the experimental cryo-EM density map, and performed in silico modeling to produce a realistic model of the complex.

Previously published WTX mutagenesis data suggest that two positively charged residues located in the tip of the loop II (R31 and R32) are simultaneously important for the toxin binding to α7-nAChR[24]. To incorporate the cryo-EM and mutagenesis data into in silico modeling of the α7-ECD/WTX complex, we used a multi-step procedure (see Methods and Supplementary Fig. 6), which was based on (1) molecular dynamics simulation of WTX and α7-ECD separately; (2) "ensemble" protein–protein docking; (3) post-scoring of docking results, employing: (a) quality of a fit into the cryo-EM density map; (b) requirement that the both R31 and R32 residues of WTX form specific interactions with nAChR; (c) requirement that the toxin's loop I is directed toward the expected membrane interface; and (d) general principles of protein packing at intermolecular interfaces. The application of these criteria permitted to select four similar models from the initial 36,400 docking solutions. One of these models is shown in Fig. 2c, d. In this model, the toxin molecule approaches the orthosteric ligand-binding site, located under the loop C of the primary (+) receptor subunit, from below; and the loop I of WTX can contact the headgroup region of the lipid membrane

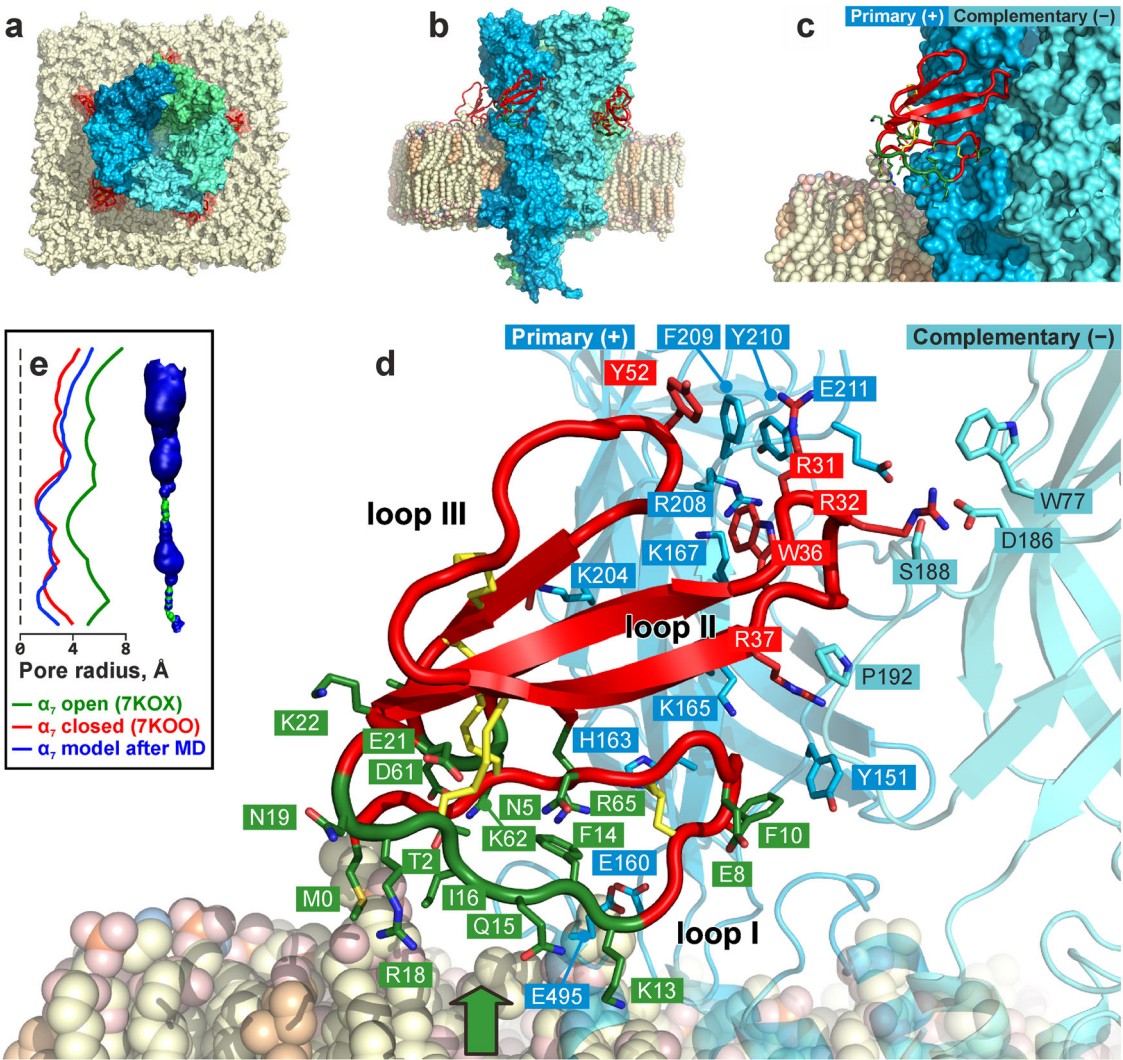

**Fig. 4 Structure of α7-nAChR/WTX complex in membrane environment. a** General view of the complex from the top. nAChR subunits are individually colored in a blue-green spectrum; five WTX molecules at the interfaces of two adjacent receptor's subunits are shown by red. **b** Side view of the complex. Carbon atoms of the membrane lipids are colored as follows: POPC and POPE, pale yellow; cholesterol, pale orange. Other lipid atoms: phosphorus, orange; oxygen, red; nitrogen, blue. Lipids of the proximal part of the membrane are removed for clarity. Membrane-interacting residues of WTX are colored by forest green. **c** Zoomed-in picture of WTX interacting by the loops II and III with α7-nAChR and by loop I, head-1, and *N*- and *C*-termini with the membrane. **d** Snapshot of WTX-nAChR and WTX-membrane interactions. α7-nAChR primary (+) and complementary (−) subunits are shown as blue and cyan cartoon, respectively; WTX residues that interact with receptor (red background) or membrane (green background) are shown. Details of the WTX membrane interactions are given in Fig. 5 (from the point of view depicted by green arrow). **e** Pore profile of the α7-nAChR channel for the given MD snapshot. Blue volume corresponds to water-permeable cavity; green—to the impermeable (e.g., closed pore). Graphs show pore radius profiles: open pore in α7-nAChR with epibatidine and PNU-120596 (PDB ID: 7KOX[16])—green; closed pore in α7-nAChR with α-Bgtx (PDB ID: 7KOO[16])—red; and the pore in our model α7-nAChR/WTX—blue.

surrounding the receptor, while the loop III touches the outer side of the loop C. The loop II of the toxin penetrates the interface between the primary (+) and complimentary (−) receptor subunits and contacts the loop C from below (Fig. 2d).

To investigate the possible interaction of the receptor-bound WTX molecule with the membrane environment of α7-nAChR, we modeled the complex of the full-length human receptor with five WTX molecules, and incorporated it into a full-atomic model of lipid bilayer (POPC:palmitoyl-oleoyl-phosphatidylethanolamine (POPE):CHOL = 2:1:1). The selected docking solution of the α7-ECD/WTX complex was used as a scaffold for the extracellular domain, and the structure of the *Torpedo* muscle-type nAChR in the closed state (PDB ID: 6UWZ[26]) was used as a scaffold for the intracellular and TM domains. We should note that in 2021 the structures of α7-nAChR in several states were

determined[16,17], but at the time of the beginning of this work they had not yet been disclosed. To check the validity of our model, we calculated the RMSD values with the recent closed-state structure of α7-nAChR (PDB ID: 7KOO[16]). Obtained values (3.0 Å for the whole structure, 2.7 Å for the ECD, 2.4 Å for the TM domain, and 2.5 Å for the ligand-binding site) were comparable with the RMSD values typically observed in MD traces of large/membrane proteins[29]. Thus, our model is precise enough to be a reliable starting point for MD calculations.

The 300 ns MD trajectory and four additional 100 ns replicas of the α7-nAChR/WTX complex provided an information about the complex stability and revealed fine dynamic details of the WTX interaction with the receptor and the membrane. The α7-nAChR/WTX complex remained relatively stable during MD replicas (see the RMSD values and lifetime of the intermolecular contacts in

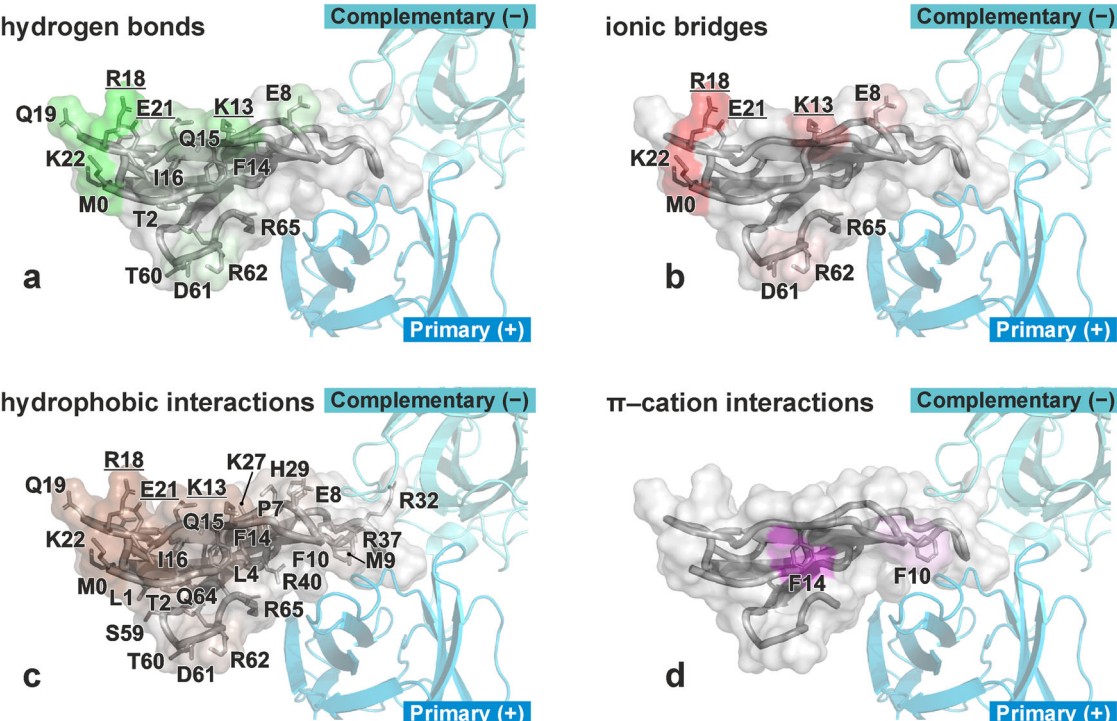

**Fig. 5 Details of toxin/lipid interactions in α7-nAChR/WTX complex in membrane environment. a–d** show WTX in surface representation viewed from "below" (from the membrane side, see green arrow in Fig. 4d), colored according to an intensity of interaction with the lipids (**a**—hydrogen bonds; **b**—ionic bridges and ion–dipole interactions; **c**—hydrophobic interactions; **d**—π–cation interactions). Primary (+) and complementary (−) subunits of α7-ECD are shown. TM-domain of α7-nAChR and lipids are omitted for clarity. WTX residues mutated in this work are underlined.

Supplementary Figs. 7 and 8, respectively), preserving main features of the starting conformation and accumulating only moderate differences in conformations and positions of five WTX molecules. Overview of the complex is given in Fig. 4a–d. Our model suggested that the R32 residue from the toxin's loop II "bridges" two adjacent nAChR subunits, forming simultaneous interactions with D186(−) (ionic bridge), S188(−) (hydrogen bond), and W77(−) (π-cation interaction) of the complementary (−) subunit and Glu211(+) (ionic bridge) of the primary (+) subunit. R31 (loop II) and Y52 (loop III) stack with the nAChR's F209(+) residue. W36 (loop II) stacks with receptor's Y210(+) and forms π-cation interactions with K167(+) and R208(+). These (and others) interactions are summarized in Supplementary Table 1.

While the loops II and III of the WTX molecule interacted with the orthosteric ligand-binding site of the receptor, the residues from the "lower" part of the toxin (loop I, N- and C-termini, and head-1 region) were in contact with the membrane surrounding α7-nAChR (Fig. 4d, shown by green). The most intensively membrane-interacting residues were K13, R18, E21, and K22, which formed a massive network of interactions with the lipids: hydrogen bonds, ionic bridges, and hydrophobic interactions (Fig. 5a–c). Two aromatic residues F10 and F14 formed π-cation interactions with the positively charged groups of the lipids (Fig. 5d). These data are summarized in Supplementary Table 2.

Analysis of the receptor's pore radius profile in the full-length α7-nAChR/WTX complex (Fig. 4e) showed that the channel pore remains in the closed conformation during MD. The calculated pore profile was similar to the profile of the closed pore observed in the cryo-EM structure of α7-nAChR in complex with α-Bgtx and dissimilar to the pore profile in the open channel in complex with agonist epibatidine and PNU-120596[16] (Fig. 4e).

**Mutagenesis revealed a crucial role of the WTX head region in the interaction with the lipid membrane and α7-nAChR.** To check the relevance of the proposed membrane active site located in the loop I and head-1 regions of WTX, we mutated two positively charged residues K13 (loop I) and R18 (head-1) to alanine. NMR titration experiments revealed that the both mutants demonstrated reduced affinity to the lipid vesicles as compared with the wild-type toxin (Fig. 3d and Supplementary Table 3). On the other hand, the decrease in the affinity for the membrane may be associated with the neutralization of the total positive charge of the WTX molecule, and therefore, the weakening of the unspecific interaction of the toxin with the anionic membrane. To avoid this, we produced the double [R18A/E21A] mutant by neutralizing two opposite charges in the head-1 region. This mutant has the same total charge (+6) as the wild-type WTX molecule, so it should demonstrate similar degree of unspecific membrane binding. NMR experiments revealed that the double [R18A/E21A] mutant demonstrated significantly lower membrane affinity compared with the single [R18A] mutant and wild-type toxin (Fig. 3d and Supplementary Table 3). Thus, the charged residues from the head-1 region are important for the WTX interaction with the membrane, and this agrees with the results of in silico modeling (Figs. 4d and 5 and Supplementary Table 2). The residual membrane affinity of the [R18A/E21A] mutant can be explained by the binding of the toxin to the membrane by the loop II.

To study the role of the membrane active site in the interaction of WTX with the full-length α7 receptor embedded into the membrane, we investigated competition of the toxin mutants with TRITC-labeled α-Bgtx at α7-nAChR expressed in HEK293 cells (Fig. 6). The binding parameters of unlabeled α-Bgtx measured in this system (IC$_{50}$ = 3.2 ± 0.4 nM, nH = 2.1 ± 0.6, Fig. 6a) were close to the $K_i$ values (0.35–3.5 nM) reported in the literature[22]. The binding parameters of WTX (IC$_{50}$ = 26 ± 4 µM, nH = 1.5 ±

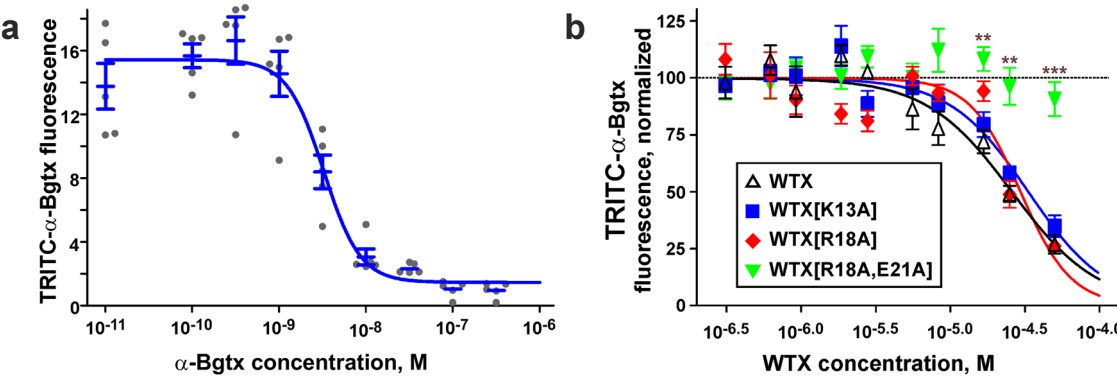

**Fig. 6 Competition of α-Bgtx and WTX and its mutants with TRITC-labeled α-Bgtx for binding to α7-nAChR expressed in HEK293 cells. a** Data on competition of α-Bgtx with TRITC-labeled α-Bgtx presented as median fluorescence intensities (MFI) ± SEM ($n = 5$ independent experiments). The Hill equation ($y = A0 + A1/(1 + ([Toxin]/IC_{50})^{nH})$) was fitted to data. **b** Data on competition of WTX and its mutants with TRITC-labeled α-Bgtx were normalized to the TRITC-α-Bgtx-binding in absence of WTX or its mutants and shown as mean ± SEM ($n = 5$ independent experiments). The Hill equation ($y = 100\%/(1 + ([Toxin]/IC_{50})^{nH})$) was fitted to normalized data. Individual datapoints for this dataset are given in Supplementary Fig. 9. **$p < 0.01$ and ***$p < 0.001$ indicate the significant difference between WTX and WTX[R18A, E21A] groups by two-sided $t$-test with Holm-Sidak correction for multiple comparisons.

0.3, Fig. 6b and Supplementary Fig. 9) were also close to the reported IC$_{50}$ values (15–47 μM)[23,24]. The single [K13A] and [R18A] mutations did not significantly change the toxin's affinity to α7-nAChR (IC$_{50}$ = 33 ± 4 and 30 ± 5 μM, respectively), while the double mutation [R18A/E21A] completely inactivated the toxin (IC$_{50}$ > 90 μM).

**Umbrella sampling simulations and calculation of the free energy changes (ΔG) confirmed important role of the membrane in the WTX interaction with α7-nAChR.** To estimate the relative contribution of the membrane in the formation of the α7-nAChR/WTX complex, we calculated binding energy (ΔG$_{bind}$) for several systems using one-dimensional potential of mean force (PMF). The toxin molecule was gradually moved from the receptor and/or membrane along the ξ direction (reaction coordinate), and umbrella sampling was performed to compute the free energy change along this pathway. ΔG$_{bind}$ was calculated separately for the WTX complex with membrane embedded α7-nAChR (Fig. 7a), for the WTX complex with extracellular domain of α7-nAChR in absence of the membrane domain and surrounding lipids (Fig. 7b), and for the WTX/membrane complex (Fig. 7c). Figure 7a–d illustrates the start and end positions of all systems under study. In the initial state, the toxin is bound to the receptor and/or membrane; in the final state, the toxin is in the free (unbound) state in the aqueous phase.

To calculate the interaction energy of WTX with α7-nAChR in the membrane, we selected from the MD trace the coordinates of the α7-nAChR/WTX complex with the largest number of the intermolecular contacts. Then we inserted the complex in the extended bilayer, and generated the intermediate WTX configurations by pulling the toxin diagonally (ξ between the X and Z axes, where Z was the membrane normal). In this case, the toxin dissociated simultaneously from the receptor and lipids (Fig. 7a). To evaluate the interaction energy of WTX with α7-nAChR without considering the membrane, we took the same α7-nAChR/WTX complex, removed the lipids and the receptor's TM and intracellular domains, and the toxin was pulled horizontally (ξ along X axis, Fig. 7b). To assess the energy of the WTX interaction with the membrane, we assembled a system with the WTX molecule located on the membrane surface in the same topology as in the complex with the receptor (Fig. 7c). In this case, the WTX molecule was pulled vertically (ξ along Z axis). The calculated one-dimensional PMF curves for all systems studied

are shown in Fig. 7e. These curves corresponded to the following binding energies: ΔG$_{\alpha7+membrane/WTX}$ = −35.8 ± 0.3 kcal/mol; ΔG$_{\alpha7-ECD/WTX}$ = −24.3 ± 0.5 kcal/mol; and ΔG$_{membrane/WTX}$ = −17.8 ± 0.4 kcal/mol.

Comparison of the calculated ΔG$_{bind}$ values showed that the interaction with the membrane add the significant contribution to the free energy of formation of the α7-nAChR/WTX complex. Interestingly, the energy contributions from the WTX interaction with the membrane and the extracellular domain of α7-nAChR are slightly non-additive. This may be due to the participation of the toxin's loop I in the interaction with the receptor and the membrane simultaneously and conformational changes of this loop during MD calculations. At the same time, due to some unknown systematic error, the calculated ΔG$_{bind}$ values were overestimated and corresponded to sub-nanomolar dissociation constants, while WTX interacts with α7-nAChR with micromolar affinity[23,24]. To confirm the systematic overestimation of the ΔG$_{bind}$ values, we performed the ΔG$_{bind}$ calculations for the α7-nAChR/α-Bgtx complex, which is characterized by nanomolar dissociation constant[22]. In the published cryo-EM structure of the α7-nAChR/α-Bgtx complex[16], the toxin does not interact with the membrane, so we took only the extracellular domain of the receptor for calculations. Starting from the cryo-EM structure and pulling the toxin in the horizontal direction (Fig. 7d), we obtained ΔG$_{\alpha7-ECD/\alpha-Bgtx}$ = −39.7 ± 0.5 kcal/mol. The difference in the calculated ΔG$_{\alpha7+membrane/WTX}$ and ΔG$_{\alpha7-ECD/\alpha-Bgtx}$ values corresponds to the difference in the dissociation constants by three orders of magnitude. This roughly agrees with the differences in the experimentally derived affinities of these toxins[22–24].

**Discussion**
Cryo-EM becomes more and more popular as the method of choice for solving various problems in structural biology due to requirement of small sample amounts and the ability to obtain high-resolution structures in a short time. However, like other methods, cryo-EM has its own limitations; and one of them is a limit on the size (mass) of a studied system[30]. In this respect, α7-ECD with the dimensions of ~9 × 9 × 6 nm$^3$ and a molecular weight ~125 kDa is close to the lower mass limit. Nevertheless, comparison of the obtained here structure of the α7-ECD/α-Bgtx complex with the structure determined previously by X-ray crystallography[15] (Fig. 1 and Supplementary Fig. 4) did not reveal significant differences. Moreover, the position of α-Bgtx in the

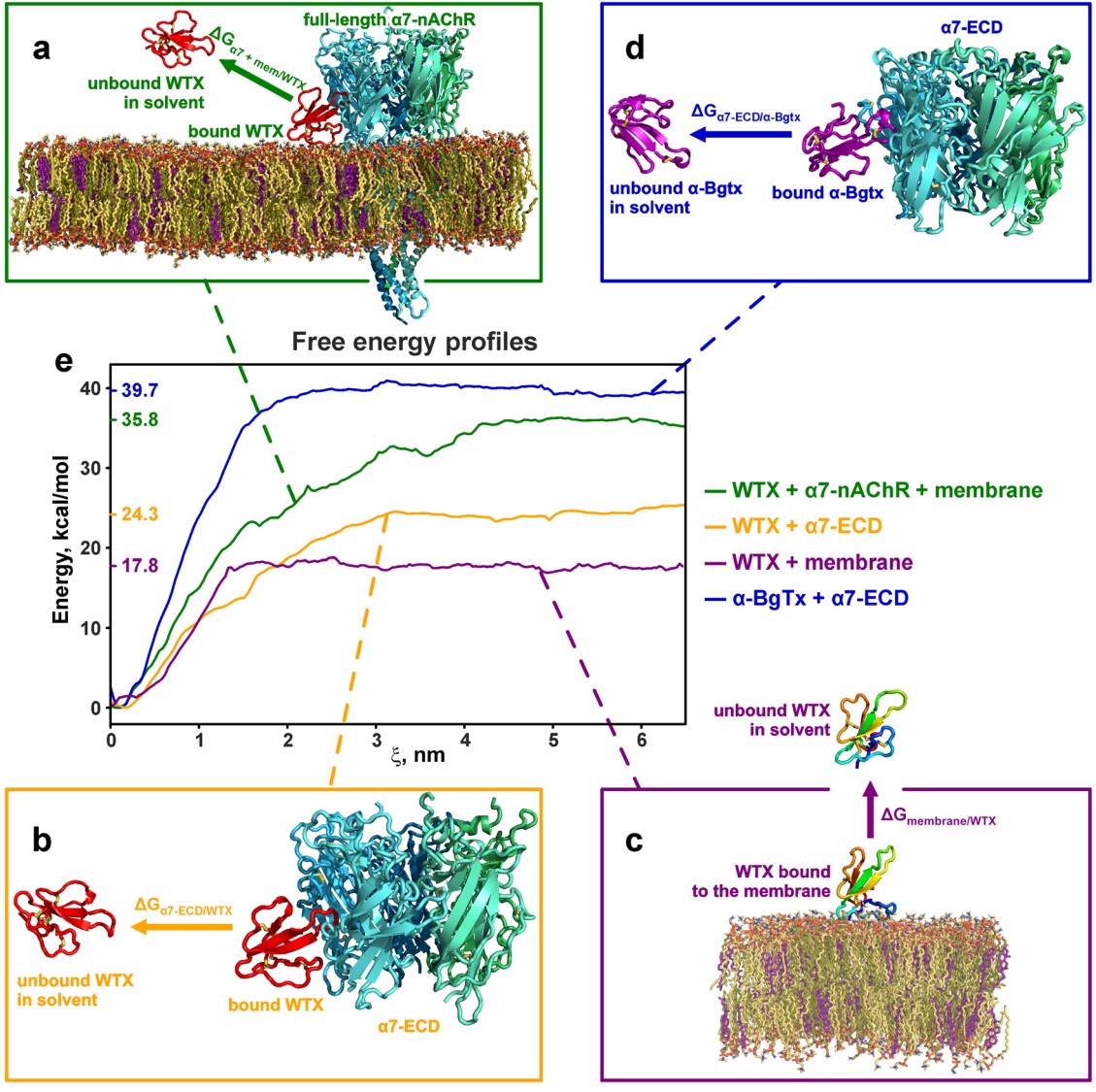

**Fig. 7 Changes of the free energy ($\Delta G_{bind}$) due to the toxins binding to α7-nAChR and/or membrane.** Start and end toxin positions for calculations of the binding energies for the complexes: **a** α7-nAChR/WTX in the membrane, **b** extracellular domain of α7-nAChR/WTX in solution, **c** membrane/WTX, **d** extracellular domain of α7-nAChR/α-Bgtx in solution. **e** Potential of mean force (PMF) curves for the studied systems.

cryo-EM structure with α7-ECD is close to that in the complex with full-length α7-nAChR[16] (Supplementary Fig. 10). Thus, chimeric α7-ECD is the good model for the structural studies of the interaction of α7-nAChR with its ligands by cryo-EM.

The relative "weakness" of the α7-ECD/WTX complex could also complicate the structural studies. Investigations of such complexes require a large excess of the ligand, which can hinder crystallization. Nevertheless, the "weak" intermolecular interactions play an important role in many essential processes involving three-finger proteins[31–34], and methods for structural studies of such complexes are in demand. Present work indicates that cryo-EM is the method of choice for studies of weak ligand-receptor complexes. In this case, excess of the ligand does not interfere with the structure determination due to its small size.

The other limitation that complicated our cryo-EM study is the problem with the preferred orientation[28,35]. Despite that α-Bgtx and WTX both belong to the family of the three-finger proteins and have similar sizes (dimensions ~ 4 × 3 × 2 nm³, MW ~8 kDa) and overall positive charge (+3 and +6, respectively), this problem was manifested differently. Thus, up to 20% of the side-view particles of the α7-ECD/α-Bgtx complex were observed on

micrographs, while only the top-view orientation of the α7-ECD/WTX complex was detected. The observed difference may be explained by the different overall shapes of the complexes (Figs. 1 and 2 and Supplementary Fig. 10). In the complex with α-Bgtx, the toxin strongly protrudes outside α7-ECD, almost doubling the particle radius. This type of interaction is achieved through the binding to α7-ECD by the elongated central loop of the toxin, while the loops I and III are just touching the receptor. In contrast, WTX with the shorter loop II is pressed against α7-ECD and interacts with the domain by the all three loops (Supplementary Table 1).

The obtained structures of the α7-ECD/α-Bgtx and α7-ECD/WTX complexes revealed substantial differences in the interaction modes of long-chain and non-conventional neurotoxins with the receptor (Supplementary Fig. 10). The loop II of α-Bgtx penetrates deeply under the receptor loop C into the orthosteric site, clamping the loop C between the loop II and prolonged C-terminal fragment of the toxin[15]. Similar modes of the α-Bgtx interaction were observed in the complexes with full-length α7-nAChR[16] and muscle type nAChR from *Torpedo californica*[26]. Interestingly, the recent structure of the complex of the

prototypical synthetic short-chain α-neurotoxin ScNtx with *Torpedo* nAChR[27] also demonstrates the similar interaction topology, where loop II of the toxin penetrates under the receptor's loop C into the orthosteric site, and the loop C of the receptor gets caught between the loops I and II of the toxin. Probably, this interaction mode is common for the complexes of long-chain and short-chain α-neurotoxins with various subtypes of nAChR. In contrast, non-conventional neurotoxin WTX pushes its loops I and II below the receptor's loop C without deep penetration into the orthosteric site, that makes its interaction mode unique. Interestingly, the contact area (~3500 Å$^2$) and number of "good" (specific) intermolecular contacts (hydrogen bonds, ionic bridges, stacking, and cation–π interactions, 18 in total) in the α7-nAChR/WTX complex exceed the same parameters for the α7-nAChR/α-Bgtx structure (~3000 Å$^2$, 6 contacts)[16]. We suggest that the configuration of the receptor/α-Bgtx complex results in a very slow toxin dissociation rate due to mutual locking of the receptor loop C and loop II of the toxin. This agrees with the significant difference in IC$_{50}$ for α-Bgtx and WTX (~3 nM and 30 μM, respectively, Fig. 6) and the almost irreversible α-Bgtx binding to the receptor[19].

The different positions of the toxins relative to the ligand-binding domain of the receptor, mentioned above, not only significantly alter the overall shape of the ECD/ligand complex, but also allow WTX to interact with the surface of the lipid membrane surrounding the α7 receptor (Fig. 4d). The membrane-active site of WTX does not demonstrate great hydrophobicity and contains many charged residues (K13, R18, E21, K22, D61, K62, and R65). They form the net of hydrogen bonds and electrostatic interactions with the lipid headgroups, including phosphate groups (Supplementary Table 2). The overall positive charge of this interface implies larger affinity of the WTX molecule to the negatively charged lipid bilayers. Indeed, we observed by NMR strong WTX partitioning to the membranes containing 10% of anionic lipids (POPG, Fig. 3). At the same time, the MD simulation of the α7-nAChR/WTX complex (Fig. 4d) and calculation of free energy of the WTX binding to the membrane (Fig. 7c) revealed that the toxin forms the stable complex with the uncharged (zwitterionic) membrane. Considering that the outer leaflet of native neuronal membranes contains a detectable fraction (~3%) of anionic lipids represented by glycolipids[36], we can assume that WTX can interact with neuronal membranes in living organisms.

Interaction with the membrane can increase the ligand's local concentration and facilitate the search and recognition of the target receptor due to more efficient two-dimensional diffusion and "pre-orientation" of the ligand. Thus, the membrane can optimize ligand-receptor interactions through the so-called "membrane catalysis" mechanism[37,38]. Previously such membrane-mediated receptor action was described for the spider gating modifier toxins interacting with the voltage-gated ion channels from the membrane-bound state[39–41]. Interestingly, not only the membrane influences the conformation and orientation of toxins, but vice versa, gating modifier toxins can alter the properties of the membrane near the receptor[42]. For example, recently it was shown that the spider toxin GsMTx-4 affects the nAChR conformation and activity by interaction with the lipid environment of the receptor[43]. The complete inactivation of WTX observed upon the [R18A/E21A] double mutation in the membrane binding site (Fig. 6b) and the changes of the free energy upon WTX binding to membrane-embedded α7-nAChR (Fig. 7) highlight the extremal importance of the membrane for the toxin-receptor interaction. Indeed, the membrane binding is responsible for ~40% of the total free energy of the α7-nAChR/WTX complex formation (Fig. 7).

Sequence analysis of snake α-neurotoxins presented in the UniProt database showed that majority of non-conventional neurotoxins (40 out of 52, 77%) have at least three positively charged residues in the region that corresponds to the membrane-active site of WTX (Supplementary Fig. 11). Thus, the "membrane catalysis" mechanism is probably common to non-conventional three-finger neurotoxins (at least to the orphan group II to which WTX belongs). In contrast, only ten of 88 long-chain neurotoxins (11%) and 21 of 103 short-chain neurotoxins (20%) have three or more positively charged residues in this region. This indicates that the membrane active site is not common among the long-chain and short-chain toxins. Interaction of the short-chain neurotoxin NTII from *Naja oxiana* (UniProt ID: P01427), which has three positively charged residues in the corresponding region, with model lipid membranes was demonstrated previously[44]. Despite this, the recent cryo-EM structure of the prototypical short-chain neurotoxin ScNtx, a homolog of NTII, in complex with muscle-type nAChR[27], like the earlier complexes of long-chain α-Bgtx[16,26], did not reveal contacts with the membrane environment of the receptor. Thus, the presence of a membrane-binding site in the toxin molecule and interaction with the membrane in the absence of a receptor does not obligatorily indicate the presence of contacts with the lipid bilayer in the toxin-nAChR complex.

The large contact area in the α7-nAChR/WTX complex revealed the multipoint type of the toxin-receptor interaction. The similar multipoint mode of interactions was previously predicted for several human three-finger proteins, which, like WTX, inhibit nAChRs in the micromolar range[31,32,45,46]. These endogenous Ly6/uPAR proteins regulate many essential functions, including neuroplasticity, embryogenesis, epithelium homeostasis, and immune responses, and can be GPI-anchored to the cell membrane (Lynx1, Lynx2, Lypd6, Lypd6B, Ly6H, etc.) or secreted (SLURP-1, SLURP-2)[18,47,48]. Additional fifth disulfide in the loop I of endogenous three-finger proteins also highlights their similarity with non-conventional neurotoxins and WTX (Supplementary Fig. 1). Probably, the weak affinity of endogenous Ly6/uPAR proteins to nAChRs is compensated by their anchoring in the membrane near the receptors. Indeed, the colocalization with nAChRs in the brain and neurons was reported for the Lynx1 and Lypd6 proteins[31,49]. The membrane-active site in the WTX molecule can serve for the same purpose. The orientation of the membrane-bound WTX molecule with the *N*- and *C*-terminal fragments located close to the surface of the lipid membrane further supports the similarity with the GPI-anchored three-finger proteins, where the GPI-anchor is attached to the *C*-terminus[18].

Another similar feature of WTX and human three-finger allosteric modulators (Lynx1, SLURP-1, SLURP-2, and Lypd6) is binding to the receptor at the site slightly different from the orthosteric site[31,32,45,46]. Comparison of the α7-nAChR/WTX and α7-nAChR/epibatidine[16] structures revealed only slight overlap between the binding sites of the toxin and agonist, which implies the possibility of simultaneous interaction of these ligands with the receptor (Supplementary Fig. 12). Moreover, similarly to Lynx1 and SLURP-2 proteins[32,50], WTX is the allosteric modulator of muscarinic acetylcholine receptors[51,52]. Altogether, this allows us to consider WTX as a model for studies of endogenous Ly6/uPAR proteins, which target nAChR.

It has been hypothesized that snake neurotoxins have been evolved from endogenous Ly6/uPAR proteins (prototoxins), retaining a homologous structural organization, but acquiring toxic properties against prey receptors[47,53]. In this context, the non-conventional neurotoxins that have a plesiomorphic pattern of five disulfide bonds with a fifth disulfide in the loop I can be considered as one of the ancient groups of neurotoxins. At the

same time, the multiple sequence alignment of cobra three-finger proteins (*Naja naja* and *Naja kaouthia* proteins presented in the UniProt database were analyzed) did not reveal a large sequence similarity of the LU-domains between non-conventional neurotoxins and non-toxic snake Ly6/uPAR proteins (average similarity between two groups ~24%). For example, WTX demonstrated the largest similarity (30%) with the Lynx2 (Lypd1) protein from *Naja naja*. On the other hand, the similarity between different groups of the toxins was slightly larger (~35% in average). The largest similarity with WTX (44%) was observed for the muscarinic toxin-like protein 2 from *Naja kaouthia* venom. Thus, the sequence analysis did not reveal a correlation between snake toxins and snake non-toxic three-finger proteins.

In conclusion, we described the structure of the complex of nAChR with the non-conventional three-finger neurotoxin. The differences in the interaction mode of long-chain, short-chain, and non-conventional three-finger neurotoxins with the nicotinic receptors were revealed. The binding to phospholipid membrane significantly enhances the interaction of the non-conventional toxin WTX with α7-nAChR. Our study provides evidence that membrane catalysis mechanism is involved in the action of some three-finger neurotoxins.

## Methods
**Recombinant production of neurotoxins and α7-ECD**. Recombinant analog of WTX, its mutants, and [15]N-labeled WTX were produced in *E. coli* as described earlier[52,54]. Correct folding of the toxin and its mutants was confirmed by 1D [1]H NMR spectroscopy. Chimera of the α7-nAChR extracellular domain with acetylcholine-binding protein (α7-ECD) containing His₆-tag at the *C*-terminus was produced in *Pichia pastoris* similarly to[14]. Briefly, the *α7-ECD* gene was cloned into the vector pPICZαA, and the domain was expressed in KM71H *Pichia pastoris* cells. The domain was purified from the growth culture on Ni²⁺-Sepharose (GE Healthcare), treated by PNGase F (New England Biolabs, cat. #P0704S), and a pentameric fraction of α7-ECD was further purified by size-exclusion chromatography (SEC) on Superdex-200 10/300 column (GE Healthcare) in buffer 10 mM HEPES, 150 mM NaCl, 3 mM EDTA, pH 7.4 20 mM Tris-HCl, 150 mM NaCl, pH 7.5 (Supplementary Fig. 13a). Ability of α7-ECD to bind α-Bgtx (Tocris, USA) at nanomolar range ($K_d$ ~26 nM) was confirmed using a surface plasmon resonance (SPR) technique (Supplementary Fig. 13b).

**SPR analysis**. Experiments were performed on SPR biosensor Biacore 8k (GE Healthcare, USA) at 25 °C using standard CM5 optical chips (GE Healthcare). Intermolecular interactions were recorded as sensorgrams representing a change of a biosensor signal in resonance units (RU). 1 RU corresponds to 1 pg of a protein per 1 mm² of a chip surface. α-Bgtx was immobilized on the SPR chip by the standard protocol. Briefly, carboxyl groups on the chip were activated by injection of a 1-ethyl-3-(3-dimethylaminopropyl)-carbodiimide-HCl and N-hydroxysuccinimide mixture (1:1) for 7 min (flow rate 5 μl/min). Then the SPR chip was washed with 10 mM sodium acetate, pH 4.5, containing 15 μg/ml of the ligands for 10 min (flow rate 2 μl/min). Unreacted activated carboxyl groups were blocked by 1.0 M ethanolamine-HCl. Analysis of the α7-ECD/α-Bgtx interaction was performed for the domain concentration in a 10–2500 nM range in 150 mM NaCl, 3 mM EDTA, 0.05% P-20 surfactant, 10 mM HEPES, pH 7.4 at 25 °C for 7 min (flow rate 5 μl/min). Biosensor channel without immobilized ligand was used as a control. The chip surface was regenerated by injection of 10 mM glycine-HCl, pH 2.0 for 0.5 min. The resulting sensorgrams were the difference between the biosensor and control signals. Rate constants ($k_{on}$, $k_{off}$) and equilibrium dissociation constant ($K_d$) were calculated using the BIA evaluation program v.4.1 (GE Healthcare).

**Cryo-EM sample preparation and image processing**. EM grids were plasma-treated using glow discharge cleaning system (PELCO easiGlow) operated under 0.26 mbar pressure of atmospheric gas. Quantifoil R1.2/1.3 Cu and Quantifoil R1.2/1.3 Au grids were glow-discharged at 20 mA for 20 s, and Quantifoil R1.2/1.3 Au grids coated with graphene oxide support film were glow-discharged at 15 mA for 5 s. EM grids coated with graphene oxide were used for automated data collection due to high stability and positive influence on a particle orientation. α7-ECD (0.15 mg/ml) purified by SEC was mixed with α-Bgtx or WTX to a final molar ratio of 1:5 (monomer of the domain to toxin). Samples with the α7-ECD/α-Bgtx or α7-ECD/WTX mixtures (3 μl) was applied to the glow discharged EM grids and immediately blotted for 2.5 s at 100% humidity and 4 °C, then plunge-frozen into liquid ethane cooled by liquid nitrogen using Vitrobot Mark IV (Thermo Fisher Scientific, USA).

Cryo-EM study was carried out with Titan Krios 60–300 TEM/STEM microscope (Thermo Fisher Scientific), equipped with XFEG electron source, Falcon II TEM direct electron detector and Cs image corrector (CEOS, Germany) at accelerating voltage of 300 kV.

Data collection for the α7-ECD/α-Bgtx complexes was performed in automated data acquisition mode using EPU software. In total, 2245 movies were collected using ×75k magnification (pixel size 0.86 Å) with the defocus values from 0.6 to 2.5 μm. Each movie was collected for 2 s and contained 40 frames. The accumulated total dose was ~80 e/Å². Motion correction, CTF estimation and particle picking were performed with Warp[55]. After preprocessing 1,517,534 particles were exported to CryoSparc[56] for the further analysis. 2D classification revealed 282,948 "side-view" particles, which were used for ab initio reconstruction and subsequent processing with Relion3[57]. 3D refinement using C5 symmetry provided the cryo-EM density map with 4.02 Å resolution. After subsequent 3D classification without alignment, 27,813 particles from the best class were chosen for the final 3D refinement. Global resolution of the obtained 3D map was estimated by the "gold-standard" criterion to be 3.37 Å. Local resolution was estimated with ResMap[58]. Attempts to combine "top-view" and "side-view" particles increased an anisotropy of the cryo-EM density map and blurring of details.

Data collection for the α7-ECD/WTX complexes was performed at 0°, 15° and 30° tilt angles. 1981 movies (862 movies at 0°, 613 movies at 15°, and 506 movies at 30°) were collected using ×75k magnification (pixel size 0.86 Å) with the defocus values from 1 to 3 μm in automated data acquisition mode using EPU software. Each movie was collected for 1.5 s and contained 30 frames. The accumulated total dose was ~80 e/Å². Motion correction, CTF estimation and particle picking were performed with Warp[55]. After preprocessing, 703,453 particles were exported to CryoSparc[56] for the further analysis. After 2D classification, 108,680 particles from the merged dataset were used for the final 3D refinement using C5 symmetry. The α7-ECD crystal structure (PDB: 4HQP[15]) without α-Bgtx filtered down to 40 Å was used as a reference.

**WTX binding to lipid vesicles**. All lipids were from Avanti Polar Lipids (Alabaster, AL). Small unilamellar vesicles were prepared using Avanti Mini-Extruder and 100 nm polycarbonate membranes. The POPC:POPG:CHOL mixture (7:1:2, mol/mol) in 20 mM MES (pH 6.5) with 50 mM NaCl was used. The final lipid concentration was monitored by 1D [1]H NMR spectroscopy by dissolving the small fractions of the vesicle preparation in the CDCl₃/CD₃OD/D₂O (15:10:3) mixture. Aliquots of the vesicle sample were added stepwise to the 35 μM samples of WTX or its mutants (5% D₂O; same buffer), and 1D [1]H NMR spectra were measured at 30 °C on a Bruker Avance 800 spectrometer equipped with a cryoprobe. The equilibrium concentration of the free protein in solution ($Cf$) was determined using the integral intensity of the amide-aromatic region of the spectrum, assuming that the bound protein is unobservable by NMR. The binding isotherms were analyzed using Langmuir equation as described in ref.[59]. The effect of dilution was accounted for.

**In silico modeling**. Reconstruction of the complex of full-length α7-nAChR (with truncated intracellular loop) with WTX in the membrane was done by the multi-step computational procedure based on the obtained cryo-EM density map of the complex, ensemble protein–protein docking, post-scoring of the docking solutions, homology modeling and MD simulations. The all-atom force field CHARMM36, which proved to correctly describe the protein dynamics[60], and the TIP3P water model recommended for it, were used for MD calculations. All MD calculations were performed using the GROMACS[61] software. The systems setup for all the performed MD calculations (including number of lipids, ions, water molecules and box dimensions) are summarized in Supplementary Table 4. The following versions of software packages were used during the work: GROMACS 2020-2022 (different versions); ZDOCK 3.0.2; IMPULSE 21.09 and PLATINUM 1.0.7. The whole bunch of molecular modeling operations included the following steps (Supplementary Fig. 6):

(1) Generation of MD-derived ensembles of conformations for WTX[P33A] and α7-ECD.

- WTX molecule demonstrates conformational heterogeneity in solution due to *cis-trans* isomerization of the Arg32-Pro33 bond[52]. Therefore, for the model building, we used the NMR structure of the WTX[P33A] mutant (PDB ID: 2MJ0[52]) corresponding to the *trans*-conformer. For this structure, GROMACS[61] "gmx cluster" routine (clustering method: gromos; cut-off: 0.25 nm) produced a set of four distinct conformations, which were used as starting points for four MD calculations of WTX in water box (trajectory length: 200 ns each; counterions were added for electroneutrality; MD step: 2 fs; pressure: 1 atm; temperature: 310 K; 10⁴ steps of steepest descent minimization were carried out before production runs). These trajectories were concatenated ("gmx trjcat" routine) into a single 800-ns dataset, and the second round of clustering with the same parameters yielded the ensemble of 14 conformations, which were used in subsequent docking runs. These conformations varied most significantly in the region of the loop II.

- The structure of α7-ECD chimera with AChBP (PDB ID: 4HQP[15]) was used to build the model of α7-ECD/WTX complex. 200-ns MD trajectory of the α7-ECD homopentameric protein in water was calculated starting from the crystal structure[15]; $10^4$ steps of steepest descent minimization were carried out before production run. After that, the whole pentameric complex was divided into five sub-trajectories containing couples of the adjacent subunits, which represent separate ligand-binding sites (protomers 1-2, 2-3, 3-4, 4-5, and 5-1). These trajectories were superimposed ("gmx trjconv -fit") and concatenated ("gmx trjcat") into a single 1000-ns dataset, which was clustered ("gmx cluster" with same settings) over the orthosteric ligand-binding site (residues 40-56, 55-70, 110-121, 148-152, 161-183, 204-225 of the primary (+) subunit and residues 52-63, 74-82, 125-146, 176-201 of the complementary (−) subunit) to yield the ensemble of 26 conformations, which were used in subsequent docking runs. These conformations varied most significantly in the region of the C-loop of the primary subunit.

(2) The obtained ensembles of WTX and α7-ECD conformations were used for the "ensemble" protein–protein docking[24,32] in the ZDOCK software[62]. To avoid unrealistic complexes, we "blocked" fragments of both molecules that a priori should not contact: α7-ECD residues other than those belonging to the orthosteric ligand-binding site (see previous paragraph) and WTX residues not belonging to the loops ("fingers"): 0–4, 14–25, 41–47, 57–62. To sample all combinations of the ligand and receptor conformations, $14 \times 26 = 364$ docking runs were performed, yielding 36400 models of the α7-ECD/WTX complex (100 solutions from each run).

(3) Post-scoring of the docking solutions (#1) was divided into several steps resulting in the gradual decrease in the number of acceptable models (Supplementary Fig. 6):

  - The first criterion was the compliance of the docking solution to the experimental 3D cryo-EM map of α7-ECD/WTX complex (Fig. 2b). The homopentameric α7-ECD/WTX complexes (containing five receptor subunits and five WTX molecules in each of the binding sites) were reconstructed from the complexes of "dimeric" ligand-binding sites with one WTX molecule obtained by docking. Each of the α7-ECD/WTX complexes was fitted to the 3D cryo-EM map using the 100-step "fitmap" procedure in UCSF Chimera[63]. Models with "correlation" >0.83 were selected.

  - Previously we identified that the R31 and R32 residues of WTX are critical for the α7-nAChR binding[24]. Therefore, it was required that R31 and R32 simultaneously form the specific intermolecular contacts (hydrogen bonds, ionic bridges, stacking, or cation–π interactions) with the receptor.

  - Finally, to satisfy the general principles of protein packing at intermolecular interfaces[64,65], we also required that: (1) the WTX surface area buried at the receptor interface should be ≥350 Å$^2$; (2) "complementarity" of the hydrophobic properties in the binding site should be ≥0.55 (measured as a molecular hydrophobicity potential, MHP[64]); (3) number of "good" (specific) intermolecular contacts (hydrogen bonds, ionic bridges, stacking, and cation–π interactions) should be ≥10. All these parameters were evaluated by the PLATINUM software[65]. After this step, we got 426 acceptable solutions of the complex.

(4) Post-scoring of the docking solutions (#2). α7-ECD has overall toroidal shape without large protruding features. As a result, α7-ECD can be fitted to the experimental cryo-EM map in various orientations, and the rotation of the protein around C5 symmetry axis is loosely defined. In this case, the overall fitting of the α7-ECD/WTX complexes was determined by the toxin density and straightforward fitting procedure (like one used at the previous step) did not discriminate between the complexes with correct and wrong α7-ECD orientations (rotation around C5 axis). The penalty function values had minor differences. At the same time, an attachment of glycan fragments to N66 of α7-ECD resulted in the prominent bulges in the cryo-EM map (Fig. 2b), thus providing the way to restrict the α7-ECD orientation.

  - To make the structure of the complex symmetrical, we duplicated the subunit with the largest glycosylation site in the PyMOL program. α7-ECD was positioned within the experimental cryo-EM map so that the glycan fragments occupied corresponding density bulges. For each of 426 models we took structures of five toxin molecules (without α7-ECD) and applied the transformation to them so that they were oriented as the pre-oriented α7-ECD. After that, the structures of α7-ECD and five toxins were merged and fitted to the EM map in UCSF Chimera using the "fit #3 #0 resolution 8 metric cam" command. This command calculated a density map for the model structure with a resolution of 8 Å, and it was fitted to the experimental density map using a "cam metric" (correlation about mean). The use of this metric made it possible to take into account protruding atoms and the degree of filling of one density with another. We took those structures with the penalty function values ("correlation about mean") ≥0.29, and with shift from the initial position during the

fitting <5 Å and rotation from the initial position during the fitting <5°. After this step, we got 21 acceptable solutions of the complex.

  - Obtained solutions were visually inspected for the filling of corresponding density by the toxin molecule. Seven and eight models were excluded from the set, where WTX did not fill the entire density or some fragments of the toxin protruded from the density, respectively. As a result, we got the two clusters of the structures. The first cluster included four similar structures in which the WTX loop I was directed toward the membrane. The second cluster included two structures in which the WTX loop III was directed toward the membrane.

  - Based on the NMR data on the membrane-binding site in the WTX molecule, the structures from the second cluster were discarded, and one structure from the first cluster was taken as a representative solution (Fig. 2d).

(5) MD of the full α7-nAChR complex with WTX in the membrane. Model of the α7-nAChR was built by homology in the MODELLER software[66]. Two structural templates were used: the model of the α7-ECD/WTX complex (see previous paragraph) for the extracellular domain, and the α-subunit of muscle-type Torpedo nAChR with the closed pore (PDB ID: 6UWZ[26]) for the transmembrane domain. Due to lack of resolved protein parts, we removed from the final model the N-terminus (residues 1–23, UniProt numbering) and TM3-TM4 cytoplasmic loop (residues 350–425). An α7-nAChR pentamer model with five WTX molecules at all binding sites was embedded in a pre-equilibrated lipid bilayer with the composition similar to the neuronal cell membrane (POPC:POPE:CHOL = 2:1:1) using the IMPULSE software[67]. Initially, the membrane fragment contained 576 POPC, 288 POPE, and 288 cholesterol molecules, but during insertion, some lipid molecules were removed to avoid overlap (see Supplementary Table 4). After addition of TIP3P water molecules and sodium ions (to maintain electroneutrality), the system was equilibrated in several stages: (1) $5 \times 10^4$ steps of steepest descent minimization; (2) heating from 5 to 310 K during 100 ps MD run; (3) 5 ns MD run at 310 K with fixed positions of the protein atoms to permit a membrane relaxation after insertion of the large transmembrane protein; (4) 50 ns MD run using Berendsen barostat to equilibrate the complex system. Finally, the 300 ns production MD run was calculated with Parrinello-Rahman barostat in the GROMACS software using the CHARMM36 force field. To increase the coverage of structural statistics, four additional 100 ns MD replicas were calculated starting from the structures at 0, 100, 200 and 300 ns of the original MD by generation of new Maxwell velocities and 2 ns equilibration.

(6) Analysis of WTX–receptor and WTX–membrane interactions. Time-resolved contacts of several types (hydrophobic, ionic, ion–dipole, and cation–π interactions, hydrogen bonds, and stacking) were determined using the IMPULSE software[67]. The whole production run trajectories were used for analysis. The key persistent WTX–receptor contacts were observed in all five MD replicas. Moreover, a strongly correlated distributions of the contact lifetimes in the different replicas were observed (Supplementary Fig. 8). Pore radius profile of the α7-nAChR channel was computed in the HOLE program[68].

**Binding free energy ($\Delta G_{bind}$) calculations**. To determine binding free energies, umbrella sampling technique was used. $\Delta G_{bind}$ values were calculated for the following systems: α7-nAChR in the membrane and WTX, the extracellular domain of α7-nAChR and α-Bgtx, the extracellular domain of α7-nAChR and WTX, membrane and WTX. Flowchart of calculations involved several steps: construction of the system, energy minimization, heating and relaxation, generation of the intermediate toxin positions via pulling, MD calculations for each intermediate state, and PMF profile[69] calculation. As a convergence criterion, we made sure that the PMF curve has reached a plateau at a sufficient distance of the toxin from the receptor or membrane. The parameters of MD calculations are collected in Supplementary Table 4.

To construct the complex of α7-nAChR in the membrane with WTX, we selected a frame from the 300 ns MD trajectory (86.05 ns) of the α7-nAChR/WTX complex, where one of five toxin molecules formed the largest number of contacts with α7-nAChR. Complex was oriented by aligning the C5 symmetry axis of α7-nAChR along the Z axis, the bound WTX molecule was oriented for pulling from the receptor along the X axis. The system was inserted into the bilayer with extended dimensions, solvated, and ions were added using the IMPULSE software. The final dimensions of the simulation box were $24.1 \times 12.1 \times 23.6$ nm. After energy minimization, the system was heated (from 5 to 310 K during 40 ps) and then relaxed during 50 ns using positional restraints on the WTX backbone atoms (300 kJ mol$^{-1}$ nm$^{-2}$). Berendsen barostat for pressure coupling was utilized for heating, relaxation, pulling and umbrella sampling runs.

Distance between the centers of masses of two α7-nAChR subunits and WTX was considered as "reaction coordinate" (ξ) for the PMF calculations[69]. The starting ξ value corresponding to the initial position of WTX and α7-nAChR was set to zero. Initial configurations for umbrella sampling were generated by running of MD simulation with timestep of 2 fs, applying constant speed of

5 nm/ns along the ξ direction (x, y, z) = (−1, 0, 0.6) during 1717 ps (until the distance between the pulled groups exceeds the half of the box size in the X direction), and saving the conformations each 2 ps. To keep the receptor subunits in their original place, the special positional restrains were imposed on the primary and complementary α7 subunits. To allow adaptation of the ligand-binding site conformation during the toxin pulling, the restraints were imposed on the residuesremote from the C-loop and the binding site (namely, Cα atoms of the residues 51–65 and 72–84 of the primary subunit and 163–169 and 218–226 of the complementary subunit (UNIPROT residue numbering)). 76 Conformations were selected as the starting structures for the umbrella sampling run. The selection procedure was following: minimum distance was rounded to nearest multiple of 0.01 nm, desired distances were 20 distances from the rounded minimum with the step of 0.05 nm, $\xi \in$ [0 nm, 1 nm]; then 10 distances with the step of 0.1 nm, $\xi \in$ [1 nm, 2 nm]; and 43 distances with the step of 0.15 nm, $\xi \geq 2$ nm. The conformations with the closest distances to the desired ones were chosen. NPT relaxation and umbrella sampling simulations were performed for the all selected starting structures during 0.3 and 1 ns, respectively, using V-rescale thermostat (310 K) and semi-isotropic Berendsen barostat (1 atm = 1.01325 bar), saving coordinates and force every 10 ps. During simulations, the reaction coordinates were restrained at the starting positions with harmonic potential with force constant 1000 kJ mol$^{-1}$ nm$^{-2}$. All other MD parameters were the same as in the MD simulation of the α7-nAChR/WTX complex. Umbrella sampling 1 ns trajectories were analyzed using the weighted histogram analysis method (WHAM) to calculate the PMF profile along the reaction coordinate[69]. $\Delta G_{bind}$ mean value and standard deviation for the α7-nAChR + membrane/WTX system was calculated by averaging the PMF plateau values at $\xi > 4.2$ nm. Zero PMF value has been set to the minimum of the curve.

To calculate $\Delta G_{bind}$ of α-Bgtx to α7-nAChR, we used the cryo-EM structure of the α7-nAChR/α-Bgtx complex (PDB ID 7KOO[16]). Using this structure, we constructed the complex of the pentameric extracellular domain of α7-nAChR (α7-ECD, residues 24–229) with one α-Bgtx molecule. The TM and intracellular receptor domains were removed in the PyMOL package. Obtained complex was placed at the edge of an elongated simulation box (24 × 12 × 12 nm$^3$), so that the receptor channel was aligned along the Z axis, and the toxin molecule was aligned along the X axis. Then the complex was solvated, and Na$^+$ and Cl$^-$ ions were added up to [NaCl] = 0.15 M and taking into account the box charge neutralization. After energy minimization, the system was relaxed during 5 ns using the positional restraints on the protein non-hydrogen atoms (1000 kJ mol$^{-1}$ nm$^{-2}$). Initial configurations for the umbrella sampling were generated by the running MD simulation with the pulling the α-Bgtx molecule along the X axis direction at a speed of 5 nm/ns. Similar to the α7-nAChR/WTX system, special restraints were imposed to maintain the receptor integrity during the pulling. For 65 selected starting α-Bgtx positions, NPT relaxation (during 0.3 ns) and the umbrella sampling MD calculations (during 2 ns) were conducted.

To calculate $\Delta G_{bind}$ of WTX to the extracellular domain of α7-nAChR, the receptor subunits were truncated to include only the extracellular domain (residues 24–229). All other procedures were identical to those used to calculate of $\Delta G_{bind}$ of the α7-ECD/α-Bgtx system.

To calculate $\Delta G_{bind}$ of WTX binding to the membrane via loop I, we took the same WTX coordinates that were used to calculate $\Delta G_{bind}$ of the α7-nAChR/WTX complex. The toxin was inserted into the bilayer, solvated, and ions were added using the IMPULSE software. The final dimensions of the simulation box were 10.8 × 10.8 × 21.7 nm$^3$. After energy minimization, the system was relaxed during 20 ns with the imposition of an additional limiting harmonic angular potential which restricted the angle between the vector connecting the loop I and III of WTX and bilayer normal (Z axis). Residues 0–21 of WTX were used to set the vector reference point in the loop I, and residues 44–65 were used to set the reference point in the loop III. The following parameters were used: pull-coord1-geometry = angle-axis, pull-coord1-rate = 0 and pull-coord1-k = 500 kJ mol$^{-1}$ rad$^{-2}$. This angular potential prevented a change of the orientation of the WTX molecule relative to the membrane during system equilibration. This was necessary to determine the energy contribution of the loop I to the interaction of WTX with the membrane. After equilibration, the WTX molecule was pulled from the membrane along the Z axis at a speed of 5 nm/ns. Then, 60 starting positions for the umbrella sampling calculations were selected as described above for the α7-nAChR/WTX system. NPT relaxation (0.3 ns) and umbrella sampling simulations (1 ns) were performed for the all selected starting structures. All other procedures were identical to those used to calculate of $\Delta G_{bind}$ of the α7-nAChR/WTX system.

**Expression of α7-nAChRs in HEK293 cells.** HEK293 cells (ATCC, USA) were cultivated in DME medium (PanEco, Russia) supplemented with 10% fetal calf serum (Cytiva, USA) and 2 mM glutamine (PanEco, Russia). For expression of functional α7-nAChR, cells were seeded at density 1 × 10$^6$ cells/ml and co-transfected with plasmids pVAX/α7-nAChR and pVAX/NACHO. NACHO is a chaperon required for the proper α7-nAChR folding and incorporation into the cell membrane[70]. Molar ratio of the pVAX/α7-nAChR and pVAX/NACHO plasmids was 1:3, respectively. The transfection was carried out using GeneJector liposome-based transfection reagent[71]. Transfected cells were cultivated for 48 h,

collected by centrifugation, fixed in 4% PFA for 1 h at 37 °C, and immediately used for competitive binding experiments. Expression of functional α7-nAChR at the cell membrane was confirmed by flow cytometry and confocal microscopy using TRITC-labeled α-Bgtx (Sigma-Aldrich, USA).

**Competition of WTX and its mutants with α-Bgtx for binding to α7-nAChR.** For competitive binding, HEK293 cells expressing α7-nAChR were incubated in PBS (pH 7.4) with 1 μM of TRITC-labeled α-Bgtx (Sigma-Aldrich,USA) and with 0.01–300 nM of α-Bgtx or with 0.3–50 μM of WTX or its mutant variants for 4 h at room temperature. After incubation, cells were washed twice and analyzed using Attune NxT flow cytometer (Lifer Technologies, USA). Minimum 5000 events in gate were collected and median fluorescence intensities were normalized to cells treated only by TRITC-labeled α-Bgtx. The gating strategy used is shown in Supplementary Fig. 14.

**Statistics and reproducibility.** Data are presented as mean ± S.E.M. Sample numbers ($n$) are indicated in the figure legends. No exclusion criteria were applied for experimental data. The data were analyzed using the one-way ANOVA with appropriate multiple comparisons post hoc test, or two-tailed $t$-test with appropriate multiple comparisons correction as indicated in the figure legends. Differences in the data were considered statistically significant at $p < 0.05$. Analysis and curve fitting were performed using the GraphPad Prism 8.0 software (GraphPad Software, San Diego, CA) or Mathematica 7 software (Wolfram Research, Champaign, IL).

**Reporting summary.** Further information on experimental design is available in the Nature Portfolio Reporting Summary linked to this Article.

## Data availability

The datasets generated and/or analyzed during the current study are available from the corresponding author on reasonable request. Source data for Figs. 3b, d and 6 are provided with the paper in Supplementary Data 1. The obtained cryo-EM data were deposited in EMDB. Accession codes: EMD-16169 (α7-ECD/α-Bgtx) and EMD-16173 (α7-ECD/WTX). The GRO files of the initial and final MD configurations, full-atom XTC trajectories utilized for data analysis, XVG files for RMSD values and contacts tables in CSV and XLSX format that support the findings of this study are available in Zenodo with the identifier: https://doi.org/10.5281/zenodo.7322819.

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

## Acknowledgements

This study made under financial support of Russian Science Foundation (project #19-74-20163). SPR measurements were performed using the equipment of "Human Proteome" Core Facility of the Institute of Biomedical Chemistry supported by MINOBRNAUKI (Agreement № 075-15-2019-1502 from 5 September 2019). Molecular modeling was done within the framework of the HSE University Basic Research Program.

## Author contributions

E.N.L. and Z.O.S. designed the study. D.S.K., M.A.S., M.V.K., A.S.P., Y.M.C., R.A.K., M.M.Z., A.O.C., M.L.B., E.O.Y., D.E.N., and A.S.I. conducted experiments and analyzed the data. E.N.L., Z.O.S., A.O.C., Y.M.C., R.A.K., A.S.P., and M.L.B. wrote the manuscript. E.N.L. and M.P.K. acquired funding and supervised the project. All authors read and approved the final manuscript.

## Competing interests

The authors declare no competing interests.
