## [Peer Review File · Communications Biology]

Reviewers' comments:

Reviewer #1 (Remarks to the Author):

The manuscript describes the interaction of beta-bungarotoxin and WTX with $\alpha 7$ -ECD. The authors have carefully designed the experiments and collected the data presented in the manuscript. They have used Cryo-EM data combined with NMR data, they define the WTX- $\alpha 7$ -ECD structure. This structure differs from canonical interaction of beta-bungarotoxin. The authors propose distinct 'membrane catalysis' mechanism for non-conventional neurotoxins.

Although most parts of the manuscript are well written, some parts require a thorough rewriting. The unusual use of words makes it difficult to follow the logic at times.

Major comments:

Although it is clear that WTX binds to phospholipids, it is important to show experimentally the interaction of the phospholipids enhances the interaction with $\alpha 7$ -ECD.

Reviewer #2 (Remarks to the Author):

Authors reported an interesting work where cryo-EM was used to study the interactions of $\alpha 7$ -nAChR extracellular ligand-binding domain ($\alpha 7$ -ECD) with its ligands, applying α -bungarotoxin (classical three-finger neurotoxin with high affinity to $\alpha 7$ -nAChR) and WTX, a non-conventional three-finger toxin which has not been widely studied. The methodology is sound and robust, and the findings are new with potential to benefit future studies involving $\alpha 7$ -nAChR as this is present in many biological systems and is a promising therapeutic target for cognitive dysfunctions and cancer therapy. The manuscript is well written and can be considered for publication following minor revision. Suggestions for improvement are given below.

Abstract Line 28: Please provide full version of WTX on its first appearance in the abstract. Suggest: The Weak Toxin (WTX) is a.....

Line 38: perhaps should be "which does not..?"

Line 68: What does author mean with "best-studied" (do you mean, most extensively studied?) - is there another word for clarity of the sentence?

Line 77 and Line 79: In the text, on the first appearance of these abbreviations (Bgtx, WTX), please spell out the full names of these toxins.

Paragraph 77-86 - Within this paragraph or before this, it would be nice to have a brief introduction about the alpha-Bgtx and WTX, in particular the source from where they are derived from (referring to updated venom proteome of which species), and their known bioactivity esp. in relation to venom action. Suggest some references as follows:

1. Alpha-BTx, typical alpha-neurotoxins from Bungarus multicinctus:
<https://www.sciencedirect.com/science/article/pii/S1532045621000909>

2. WTX: Weak toxins found in Naja kaouthia venoms of various locales:
<https://pubmed.ncbi.nlm.nih.gov/25748141/>

Line 90: subtype

Line 387: SPR - spell in first on its first appearance.

Line 471-472 : the templates of the two related proteins used in this homology-based approach - were they specific to any species?

Reviewer #3 (Remarks to the Author):

In this study, Shenkarev and colleagues studied a class of snake three-finger toxins with nicotinic acetylcholine receptors. They focus their attention to a weak non-conventional neurotoxin WTX from *Naja kaouthia*. This type of neurotoxin binds with weak affinity to mammalian nAChRs (tens μM), including the $\alpha 7$ nAChR. To validate their approach, the authors first determined the cryo-EM structure of the $\alpha 7$ -AChBP with α -BgTx and compared it to the known X-ray crystal structure of this complex. In this context it should be noted that the authors use the term " $\alpha 7$ -ECD" whereas the protein really is the $\alpha 7$ -AChBP. This set aside, the authors went on to determine the cryo-EM structure of the $\alpha 7$ -AChBP/WTX complex. Detailed molecular insight into the toxin-receptor interaction is precluded by the low resolution of the data (5.61 Angstrom). This resolution is insufficient to reveal secondary structures and this a major weakness in the study. The authors proceeded with a combination of complementary techniques to further explore the toxin-receptor interaction. These techniques include NMR spectroscopy of the free WTX molecule, combined with previous mutagenesis data and in silico docking. Despite the low resolution of the data, the authors suggest a binding mode in which the WTX toxin interacts at a site that also includes the membrane. Finally, the authors employ molecular dynamics simulations and mutagenesis data to further explore this interaction mode.

In my opinion, the low resolution of the cryo-EM for the WTX-AChBP complex precludes the detailed interpretation of the toxin binding mode. The authors engage in using a multi-method approach to reveal the correct binding mode, including mutagenesis, docking and MD simulations. However, my feeling is that the quality of the data just do not justify the conclusions made.

Reviewer #4 (Remarks to the Author):

This article implements multiple biophysical methods to elucidate the binding mechanism of the non-conventional toxin WXT to the $\alpha 7$ -nAChR receptor. The authors utilized a soluble extra cellular domain of $\alpha 7$ -nAChR for structural studies by cryoEM, NMR interaction studies of WXT with model membranes and combined these data in-silico studies to derive a $\alpha 7$ -nAChR/WTX structure consistent with the experimental data. Although the cryoEM data are low resolution the resultant models are consistent. Overall, the studies are rigorous and provide insight into the binding of non-conventional toxins to $\alpha 7$ -nAChR and merit publication after minor revisions and points to address below.

1. The authors used the muscle type structure to build the full length $\alpha 7$ -nAChR model. While a high-resolution structure of $\alpha 7$ -nAChR is now available. This is only pointed out in the methods but should be more clearly stated in the main txt so the reader is aware. A related point is the statement that the RMSD values of "3.0 Å for the whole structure, 2.7 Å for the ECD, 2.4 Å for the TM domain, and 2.5 Å for the ligand-binding site" between the homology model and the actual recent structure "indicate that our model is precise enough and is a reliable starting point for MD calculations". This reviewer is unaware of any objective RMSD criteria that indicates a model is precise enough for starting MD calculations. If there is one, then the authors should reference accordingly.

2. The lipid binding studies of the WTX mutants are limited and somewhat surprising. None of the positive charged mutations alter affinity, however the negative charge mutation combined with a single positive mutation abolishes binding. It would be most informative to include the single E21A

mutant in these studies to validate this result.

3. The study would benefit from performing the in-silico approach with the α -Bgtx complex as this structure is known. This would provide further confidence in the results from the α 7-nAChR/WTX complex.

4. The supplement would benefit from including a cryoEM data table. By convention defocus values are positive numbers and not negative as stated in the text.

5. There are several minor typographical errors that could be corrected.

Reviewer #5 (Remarks to the Author):

The manuscript from Shenkarev and coworkers shows the interaction of a weak three-finger toxin with α 7-ECD by cryomicroscopy. They show that the local interaction differs from that of α -bungarotoxin, a toxin with a high affinity to this receptor. They were able to complement this information using some mutants, RMN studies, molecular modeling, and some previously published data. The article presents some relevant new information on the binding mechanism of the WTX molecule to the α 7-nAChR especially related to the relevance of the toxin binding to the membrane.

Although I am not an expert in cryomicroscopy, I was able to apprehend the applied methodology and the conclusions reached by the authors.

Their initial motivation, as described in the introduction section, was to use the WTX as a tool to understand the interaction of molecules such as that from the Ly6/uPAR family, which can bind nAChR with low affinity. Although this is an interesting idea, one question arose: why they did not use the human three-fingers molecules themselves? I think that it would be interesting to clarify which are the common structural features, especially that related to the binding site and to the membrane-binding characteristics.

The authors stated that line 359 "Altogether, this allows to consider WTX as the prototypical neuromodulator from the toxin kingdom" It would be of interest to verify the sequence similarities of the endogenous three fingers proteins from snakes in order to consider this a plausible hypothesis.

Minor comments:

The abstract does not have any consideration on human three-fingers proteins. The introduction section should consider only the novelty of revealing other mechanisms of interaction between two correlated toxins... The discussion of human three-finger could be present only in the discussion section. Otherwise, the authors should explain better why they are not using human molecules for this study.

Results are well described, and figures are well picked and of good quality.

Methods are well described and I believe that is reproducible.

The manuscript is of interest to Toxinology and Neuroscience.

Reviewer #1 (Remarks to the Author):

The manuscript describes the interaction of beta-bungarotoxin and WTX with $\alpha 7$ -ECD. The authors have carefully designed the experiments and collected the data presented in the manuscript. They have used Cryo-EM data combined with NMR data, the define the WTX- $\alpha 7$ -ECD structure. This structure differs from canonical interaction of beta-bungarotoxin. The authors propose distinct 'membrane catalysis' mechanism for non-conventional neurotoxins.

Although most parts of the manuscript are well written, some parts require a thorough rewriting. The unusual use of words makes it difficult to follow the logic at times.

Answer:

We tried to do our best to improve the English language of our manuscript and rewrote some sections.

Reviewer #1

Although it is clear that WTX binds to phospholipids, it is important to show experimentally the interaction of the phospholipids enhances the interaction with $\alpha 7$ -ECD.

Answer:

Thank you for this valuable remark. The experimental confirmation of this idea is not easy. At present we even could not envisage the method how to do that. Especially if we have soluble variant of $\alpha 7$ -ECD without membrane interacting part. The way how to conduct such experiments on the full-length membrane-embedded receptor is also non-evident. Therefore, we used computer simulations (deltaG calculations with umbrella sampling) to show this. We calculated the energy of interaction of WTX with the full-length $\alpha 7$ receptor embedded into the membrane, with the extracellular domain of the receptor, and with the lipid bilayer in the absence of the receptor. Data obtained confirmed the significant contribution of the membrane (up to 40%) into the free energy of WTX binding to the receptor. These data were included in the revised manuscript (Figure 7).

Reviewer #2 (Remarks to the Author):

Authors reported an interesting work where cryo-EM was used to study the interactions of $\alpha 7$ -nAChR extracellular ligand-binding domain ($\alpha 7$ -ECD) with its ligands, applying α -bungarotoxin (classical three-finger neurotoxin with high affinity to $\alpha 7$ -nAChR) and WTX, a non-conventional three-finger toxin which has not been widely studied. The methodology is sound and robust, and the findings are new with potential to benefit future studies involving $\alpha 7$ -nAChR as this is present in many biological systems and is a promising therapeutic target for cognitive dysfunctions and cancer therapy. The manuscript is well written and can be considered for publication following minor revision. Suggestions for improvement are given below.

Abstract Line 28: Please provide full version of WTX on its first appearance in the abstract.
Suggest: The Weak Toxin (WTX) is a.....

Answer:

Thank you. Done.

Reviewer #2

Line 38: perhaps should be "which does not..?"

Answer:

Of cause. Thank you. Corrected.

Reviewer #2

Line 68: What does author mean with "best-studied" (do you meant, most extensively studied?) - is there another word for clarity of the sentence?

Answer:

Corrected.

Reviewer #2

Line 77 and Line 79: In the text, on the first appearance of these abbreviations (Bgtx, WTX), please spell out the full names of these toxins.

Answer:

Corrected.

Reviewer #2

Paragraph 77-86 - Within this paragraph or before this, it would be nice to have a brief introduction about the alpha-Bgtx ad WTX, in particular the source from where they are derived from (referring to updated venom proteome of which species), and their known bioactivity esp. in relation to venom action. Suggest some references as follows:

1. Alpha-BTx, typical alpha-neurotoxins from Bungarus multicinctus:
<https://www.sciencedirect.com/science/article/pii/S1532045621000909>
2. WTX: Weak toxins found in Naja kaouthia venoms of various locales:
<https://pubmed.ncbi.nlm.nih.gov/25748141/>

Answer:

Thank you for this suggestion. We added a required information.

Reviewer #2

Line 90: subtype

Answer:

Corrected.

Reviewer #2

Line 387: SPR - spell in first on its first appearance.

Answer:

Done.

Reviewer #2

Line 471-472 : the templates of the two related proteins used in this homology-based approach - were they specific to any species?

Answer:

We modelled the complex of the full-length human $\alpha 7$ -nAChR with five WTX molecules using two templates. The docking solution of the $\alpha 7$ -ECD/WTX complex was used as a template for the extracellular domain of the receptor. As the $\alpha 7$ -ECD has a 71% sequence similarity with the human $\alpha 7$ -nAChR, the template was specific to human. The structure of the muscle-type nAChR from

Torpedo californica was used as a scaffold for the intracellular and transmembrane domains. We rewrote corresponding section of the manuscript.

Reviewer #3 (Remarks to the Author):

In this study, Shenkarev and colleagues studied a class of snake three-finger toxins with nicotinic acetylcholine receptors. They focus their attention to a weak non-conventional neurotoxin WTX from *Naja kaouthia*. This type of neurotoxin binds with weak affinity to mammalian nAChRs (tens μM), including the $\alpha 7$ nAChR. To validate their approach, the authors first determined the cryo-EM structure of the $\alpha 7$ -AChBP with α -BgTx and compared it to the known X-ray crystal structure of this complex. In this context it should be noted that the authors use the term " $\alpha 7$ -ECD" whereas the protein really is the $\alpha 7$ -AChBP. This set aside, the authors went on to determine the cryo-EM structure of the $\alpha 7$ -AChBP/WTX complex. Detailed molecular insight into the toxin-receptor interaction is precluded by the low resolution of the data (5.61 Angstrom). This resolution is insufficient to reveal secondary structures and this a major weakness in the study. The authors proceeded with a combination of complementary techniques to further explore the toxin-receptor interaction. These techniques include NMR spectroscopy of the free WTX molecule, combined with previous mutagenesis data and in silico docking. Despite the low resolution of the data, the authors suggest a binding mode in which the WTX toxin interacts at a site that also includes the membrane. Finally, the authors employ molecular dynamics simulations and mutagenesis data to further explore this interaction mode.

In my opinion, the low resolution of the cryo-EM for the WTX-AChBP complex precludes the detailed interpretation of the toxin binding mode. The authors engage in using a multi-method approach to reveal the correct binding mode, including mutagenesis, docking and MD simulations. However, my feeling is that the quality of the data just do not justify the conclusions made.

Answer:

We respect the opinion of the reviewer but do not agree with it. We used the battery of the modern structural and biochemical methods to prove the binding mode of WTX with $\alpha 7$ -AChBP. Moreover, all other referees found our data and conclusions consistent and relevant.

Reviewer #4 (Remarks to the Author):

This article implements multiple biophysical methods to elucidate the binding mechanism of the no-conventional toxin WXT to the $\alpha 7$ -nAChR receptor. The authors utilized a soluble extra cellular domain of $\alpha 7$ -nAChR for structural studies by cryoEM, NMR interaction studies of WXT with model membranes and combined these data in-silico studies to derive a $\alpha 7$ -nAChR/WTX structure consistent with the experimental data. Although the cryoEM data are low resolution the resultant models are consistent. Overall, the studies are rigorous and provide insight into the binding of non-conventional toxins to $\alpha 7$ -nAChR and merit publication after minor revisions and points to address below.

1. The authors used the muscle type structure to build the full length $\alpha 7$ -nAChR model. While a high-resolution structure of $\alpha 7$ -nAChR is now available. This is only pointed out in the methods but should be more clearly stated in the main txt so the reader is aware.

Answer:

Thank you for this valuable remark. We rewrote the corresponding sections of the manuscript to include these data in the results section.

Reviewer #4:

A related point is the statement that the RMSD values of “3.0 Å for the whole structure, 2.7 Å for the ECD, 2.4 Å for the TM domain, and 2.5 Å for the ligand-binding site” between the homology model and the actual recent structure “indicate that our model is precise enough and is a reliable starting point for MD calculations”. This reviewer is unaware of any objective RMSD criteria that indicates a model is precise enough for starting MD calculations. If there is one, then the authors should reference accordingly.

Answer:

We agree with the reviewer. We rewrote the corresponding section of the manuscript and provide additional reference to the typical RMSD values observed in the MD simulations of membrane proteins.

Reviewer #4

2. The lipid binding studies of the WTX mutants are limited and somewhat surprising. None of the positive charged mutations alter affinity, however the negative charge mutation combined with a single positive mutation abolishes binding. It would be most informative to include the single E21A mutant in these studies to validate this result.

Answer:

In this work we obtained WTX and its mutants via invitro refolding of the proteins produced in the form of inclusion bodies. We controlled the refolding of each protein batch by the 1D NMR. The folded three-finger toxins have large fraction of beta-structure, and the presence of it is evident from the spectra. We tried expression of the single E21A mutant three times, but were unsuccessful in refolding of it. It should be noted that we additionally tested different protocols for the refolding, but all our attempts were unsuccessful. This is surprising, as double mutant [R18A/E21A] was refolded successfully. Probably, the overall neutral or negative charge of R18-E21 fragment is important for stabilization of the correct WTX structure, but overall positive charge in this fragment brakes down the three-finger structure.

Reviewer #4

3. The study would benefit from performing the in-silico approach with the α -Bgtx complex as this structure is known. This would provide further confidence in the results from the α 7-nAChR/WTX complex.

Answer:

Thank you for this remark. We assembled the complex of α -Bgtx with the extracellular domain of the α 7-nAChR, relaxe it through MD and calculated changes of the free energy upon the toxin binding to the receptor. The obtained data were used for comparison with the results of free energy calculations for WTX binding. We calculated the energy of interaction of WTX with the full-length α 7 receptor in the membrane, with the extracellular domain of the receptor, and with the lipid bilayer in the absence of the receptor. Data obtained confirmed the significant contribution of the membrane (up to 40%) into the free energy of WTX binding to the receptor. These data (together with results of the calculations for α -Bgtx) were included in the revised manuscript (Figure 7).

Reviewer #4

4. The supplement would benefit from including a cryoEM data table. By convention defocus values are positive numbers and not negative as stated in the text.

Answer:

Done.

Reviewer #4

5. There are several minor typographical errors that could be corrected.

Answer:

We checked the text and did our best to improve grammar and spelling.

Reviewer #5 (Remarks to the Author):

The manuscript from Shenkarev and coworkers shows the interaction of a weak three-finger toxin with $\alpha 7$ -ECD by cryomicroscopy. They show that the local interaction differs from that of α -bungarotoxin, a toxin with a high affinity to this receptor. They were able to complement this information using some mutants, RMN studies, molecular modeling, and some previously published data. The article presents some relevant new information on the binding mechanism of the WTX molecule to the $\alpha 7$ -nAChR especially related to the relevance of the toxin binding to the membrane.

Although I am not an expert in cryomicroscopy, I was able to apprehend the applied methodology and the conclusions reached by the authors. Their initial motivation, as described in the introduction section, was to use the WTX as a tool to understand the interaction of molecules such as that from the Ly6/uPAR family, which can bind nAChR with low affinity.

Although this is an interesting idea, one question arose: why they did not use the human three-fingers molecules themselves? I think that it would be interesting to clarify which are the common structural features, especially that related to the binding site and to the membrane-binding characteristics.

Answer:

We fully agree with the reviewer, that the study of the interaction of three-finger human proteins with nAChRs is very interesting and relevant task. We are working on the human three-finger proteins also and hope to be able to determine how they interact with the nicotinic receptor in the future. But the present work is devoted to investigation of non-conventional weak toxins. We for the first time determined the interaction mode of non-conventional toxins with $\alpha 7$ -nAChR and showed that it is different from the interaction modes of short-chain and long chain neurotoxins. We rewrote introduction and discussion sections of our manuscript. Now the idea to use WTX as a model of human proteins is appeared only in the end of the discussion.

Reviewer #5

The authors stated that line 359 "Altogether, this allows to consider WTX as the prototypical neuromodulator from the toxin kingdom" It would be of interest to verify the sequence similarities of the endogenous three fingers proteins from snakes in order to consider this a plausible hypothesis.

Answer:

Thank you for this interesting proposal. We took the sequences of all three-finger proteins from *Naja naja* and *Naja kaouthia* cobras presented in the UniProt database and analyze the dependences between them by multiple-sequence alignment. (The *Naja naja* proteins were used because the genome and several proteomes of this snake were published and the UniProt database contains lot of the protein entries.) Unfortunately, results of the alignment did not confirmed the

large homology between non-conventional toxins and non-toxic Ly6/uPAR proteins from snakes. We added the short description of these results in the end of the discussion section.

Reviewer #5

Minor comments:

The abstract does not have any consideration on human three-fingers proteins. The introduction section should consider only the novelty of revealing other mechanisms of interaction between two correlated toxins... The discussion of human three-finger could be present only in the discussion section. Otherwise, the authors should explain better why they are not using human molecules for this study.

Answer:

To avoid this confusion, we removed the description of the human three-finger proteins from the introduction section of the manuscript and focused the text on snake three-finger proteins. We rewrote the Discussion section also. Thank you for this remark.

Reviewer #5

Results are well described, and figures are well picked and of good quality.

Methods are well described and I believe that is reproducible.

The manuscript is of interest to Toxinology and Neuroscience.

Answer:

Thank you for such a high assessment of our work!

REVIEWERS' COMMENTS:

Reviewer #1 (Remarks to the Author):

In this revised version, the authors have added substantial amount of new data, particularly from molecular dynamics, and added further evidence to their proposal. The data suggests that WTX binding to the receptor differs from that of beta-bungarotoxin and the interaction of WTX with phospholipid membrane plays critical role in enhancing its interaction with the receptor.

The writing and language usage is substantially improved.

Reviewer #2 (Remarks to the Author):

Authors have responded to the reviewer's comments and suggestions. They have amended or modified the manuscript where appropriate. This work is interesting and it served its purpose to reveal the important role of the membrane for interaction of non-conventional neurotoxins with the nicotinic receptors, thus providing insights into the biological/toxicological activity of these lesser known toxins from snake venoms. In my opinion, the work can be accepted for publication pending some minor formatting and language check by the editorial.

Reviewer #4 (Remarks to the Author):

The authors have responded to all the major criticisms and this reviewer appreciates the additional simulation data comparing the Bgtx to WTX which increases the impact of this study. Acceptance is strongly supported.

Reviewer #5 (Remarks to the Author):

The reviewed manuscript took into consideration almost every comment of all five referees, it improved the results of the binding with the membrane by calculating Free Energy in a computation simulation using umbrella sampling simulations and the one-dimensional PMF method. By this method, they show that WTX binds strongly to the receptor in the presence of a membrane. The manuscript has better quality and is worth its publication.